# Designing Rules to Pick a Rule: Aggregation by Consistency

**Ratip Emin Berker**[1], **Ben Armstrong**[2], **Vincent Conitzer**[1,3] **& Nihar B. Shah**[1]
[1]Carnegie Mellon University, [2]Tulane University, [3]University of Oxford
{rberker, conitzer, nihars}@cs.cmu.edu, research@benarmstrong.ca

## ABSTRACT

Rank aggregation has critical applications for developing AI agents, as well as for evaluating them. However, different methods can give rise to significantly different aggregate rankings, impacting these applications. Indeed, work in social choice and statistics has produced many rank aggregation methods, each with its desirable properties, but also with its limitations. Given this trade-off, how do we decide which aggregation rule to use, *i.e.*, what is a good *rule picking rule (RPR)*? In this paper, we design a data-driven RPR that identifies the best method for each dataset without assuming a generative model. The principle behind our RPR is to maximize consistency if the data collection process was repeated. We show that our method satisfies several consistency-related axioms failed by a wide class of natural RPRs. While we prove that the computational problem of maximizing consistency is hard, we provide a sampling-based implementation that is efficient in practice. We run this implementation on known statistical models to experimentally demonstrate its desirable properties, as well as on real-world data where our method provides important insights into how to improve consistency.

## 1 INTRODUCTION

Suppose you have a collection of items, and evaluators who individually rank them. Finding the best method for aggregating such data is an age-old problem, with critical use cases in artificial intelligence and machine learning. For example, the recent surge of large language models has been driven in part by *reinforcement learning from human feedback (RLHF)* (Christiano et al., 2017; Ziegler et al., 2020), where evaluators provide ordinal preference data over model outputs, which must then be aggregated into a reward model. In other cases, human evaluators can be replaced with high-level principles, each of which ranks outputs based on how compliant they are with that principle, as is done in *constitutional AI* (Bai et al., 2022). Further, rank aggregation can be useful not just for developing capable AI models, but also for evaluating their performance. For example, interpreting incomparable benchmarks as separate evaluators, Lanctot et al. (2025a) aggregate these rankings to compare the overall performance of competing models. Last but not least, *peer review* is an essential tool for evaluating ML research, where asking reviewers to rank their assignments—later to be aggregated for final decisions—can help mitigate several shortcomings of other evaluation formats; *cf.*, for example, an experiment by Liu et al. (2022) on ICLR 2017 conference data.

In each of these use cases, different methods can lead to vastly different aggregations, thereby affecting the outcome. Failure to pick "good" methods can result in outcomes inconsistent with the rankings of the evaluators or benchmarks, as has been observed in RLHF (Ge et al., 2024; Xu et al., 2024) and agent evaluation (Lanctot et al., 2025b). But what makes an aggregation method *good*?

One common approach to this question is the *axiomatic approach* from the social choice literature (Plott, 1976): first select certain criteria (*axioms*) that the aggregation should satisfy, then design rules that meet these axioms. However, celebrated impossibility results (Arrow, 1963; Gibbard, 1973; Satterthwaite, 1975) prove some fundamental axioms are incompatible, eliminating all hope for one "ideal" rule fulfilling every desideratum. On the other hand, even if the chosen axioms are satisfiable at once, there may be many rules that do so, making the selection among them arbitrary.

Another approach for picking the aggregation method, which we refer to as the *statistical approach*, is to treat the rankings as noisy estimates of an objective ground truth. By assuming a noise model

(*e.g.*, Plackett (1975)-Luce (1959) or Mallows (1957)) for the data generation, the aggregate ranking can be chosen as the one maximizing the likelihood of the data under this model. A key challenge is the accuracy of the assumed model, which is commonly addressed by considering multiple models and choosing one via cross validation (Zucchini, 2000). However, irrespective of model selection, assuming a ground truth may be fundamentally flawed, especially in settings with legitimate differences of opinion, such as AI alignment (Ge et al., 2024). Further, many natural voting rules with desirable properties are not the maximum likelihood estimator (MLE) for *any* noise model (Conitzer & Sandholm, 2005; Conitzer et al., 2009) and are thereby precluded by the statistical approach.

Given this vast array of (partly incompatible) tools from statistics and social choice, and no clear way *a priori* of selecting from them, a natural question emerges: *In a given setting, how do we pick which rule to use? In other words, what makes a good rule picking rule (RPR)*?

Our work addresses this question. Unlike previous literature that largely focuses on picking an aggregate ranking, we want to explicitly pick a rule. Such an approach has several benefits: First, employing an RPR naturally leads to better interpretability by providing a formal justification of why other rules (under which the winners could be different) were not adopted. Second, different (perfectly reasonable) rules may be appropriate for different settings with different requirements, and RPRs offer a principled way of deciding which rule is the most appropriate for a given setting. Lastly, as we will see, our novel framework allows designing natural RPRs that can choose from *any* set of rules, making it easy to continually incorporate novel rules into the aggregation process.

**Our contributions are as follows: (1)** We introduce a novel framework for formally defining *rule picking rules (RPR)* (Section 3). Our framework allows designing principled ways of adopting a rule appropriate for the data, without committing to a set of axioms or a generative model *a priori*.

**(2)** Inspired by prior work emphasizing the link between consistency and quality in related settings, we introduce our own RPR, *Aggregation by Consistency (AbC)*, with the explicit goal of maximizing the consistency in the output if the data collection process was repeated (Section 4).

**(3)** We define several natural axioms for RPRs, and prove $AbC$ satisfies them, including those failed by a wide class of RPRs. For two axioms that $AbC$ fails, we prove impossibility results (Section 5).

**(4)** We prove that the computational problem of checking if "complete" consistency can be achieved for a given input (*i.e.*, picking a rule that produces the exact same output) is NP-complete (Section 6).

**(5)** Nevertheless, we provide an implementation of $AbC$ that is efficient in practice and performs well in experiments on known distributions (Section 7). Our implementation includes a learning component that optimizes consistency over infinitely many *positional scoring rules*. We show that $AbC$ can be applied to both score- and rank-based evaluations across a large variety of empirical settings, at times improving significantly upon the consistency given by rules used in practice.

Omitted proofs are in Appendix B. An implementation of $AbC$ was awarded as one of the four winners in the *2nd Computational Social Choice Competition* at the 33rd International Joint Conference on Artificial Intelligence, which scored rules via a hidden welfare function, demonstrating $AbC$ performs well in general settings. Taken together, our work lays a robust theoretical and computational foundation for principled rule picking, paving the way for future work in this novel framework.

## 2 OVERVIEW OF THE PROPOSED APPROACH: WHY CONSISTENCY?

Before introducing our formal framework, we discuss the basic idea of our approach. Suppose we have two sets of evaluators (*e.g*, benchmarks or human voters) who independently rank the same items (*e.g.*, AI models/outputs or academic papers). A popular measure of the "quality" of an aggregation method is the *consistency* between its output on two independent sets of rankings. As such, we seek to design an RPR that picks the rule that maximizes this consistency.[1] This intuition is inspired by a number of related settings where past work emphasizes the importance of consistency:

**(1) Peer review** is classical setting in which evaluations need to be aggregated, and much work has been done to optimize this process; *cf.* Shah (2022) for an overview. Many experiments split reviewers into two panels evaluating the same data (Jecmen et al., 2022), where interpanel disagreement

---

[1]We will restrict our RPR to rules that satisfy basic axiomatic properties such as neutrality (all items being ranked are treated equally), which avoids the pathological case of a constant function with maximal consistency.

is interpreted as a shortcoming (Obrecht et al., 2007; Fogelholm et al., 2012; Pier et al., 2017; Bast, 2020). For example, in the NeurIPS 2014 and 2021 conferences, the two panels disagreed on over half of accepted papers, taken as a sign of arbitrariness in the review process (Lawrence & Cortes, 2014; Cortes & Lawrence, 2021; Beygelzimer et al., 2023). Interpanel consistency is also used to compare distributed peer review with an expert panel (Patat et al., 2019; Kerzendorf et al., 2020). Overall, it is clear that the peer review community considers consistency an indicator for quality.

**(2) Clustering:** One can view the task of rule picking as *learning* a rule from rankings. Unlike earlier work with a ground truth rule (Procaccia et al., 2009), our setting is *unsupervised* and thus closely related to clustering. Indeed, Ailon et al. (2005) show clustering and rank aggregation can be approached with near-identical algorithms. Similar to rule picking, an important challenge in clustering is model selection (*e.g.*, number of clusters). Prior work shows picking the model maximizing stability—*i.e.*, obtains similar results on several datasets from an underlying model—yields higher accuracy (von Luxburg, 2010). Again, consistency and quality of the output are closely related.

**(3) Minimum-variance unbiased estimator (MVUE):** Among all unbiased estimators (expected value equals the true value of the parameter being estimated), the one with the smallest mean squared error has the lowest variance. Indeed, MVUEs are commonly studied in statistics (Chapman & Robbins, 1951; Rao, 1949) and related bounds can be used for rank aggregation (Hajek et al., 2014; Khetan & Oh, 2016). Consider applying an unbiased estimator to two i.i.d. datasets. As a variable's variance equals half the expected squared difference between its two i.i.d. copies, the estimator with the smallest expected difference between the two datasets is again the one with the min. variance, *i.e.*, the MVUE. This motivates the most consistent rule in our setting, where unbiasedness is interpreted as basic constraints for any acceptable rule, which we achieve by restricting our RPR to pick from neutral & anonymous rules (those that treat all items and evaluators the same, respectively).

**(4) AI Alignment:** There is nascent interest in applying tools from social choice to AI alignment processes, such as reinforcement learning from human feedback (RLHF) (Conitzer et al., 2024; Dai & Fleisig, 2024; Mishra, 2023), with particular emphasis on consistency. Much like peer review, RLHF inevitably relies on a limited set of evaluators, despite aiming for broad societal alignment. As such, aggregation methods that are robust to repetitions of the process can decrease the arbitrariness of the final AI model due to the choice of evaluators. As noted by Conitzer et al. (2024), the focus of social choice on producing consistent aggregations makes it an appropriate tool for this setting.

While these four settings (peer review, clustering, MVUEs, and RLHF) motivate a rule picking rule that maximizes the consistency between two independent evaluations, it is not clear how one would implement that. After all, in practice, we often have only one copy of the process. To explain how we circumvent this, **Algorithm 1 introduces our RPR, named *Aggregation by Consistency (AbC)*.**

---

**Algorithm 1:** Aggregation by Consistency ($AbC$), informal (see Section 4 for formal definition).

**Input:** A set of evaluations over items, a set of acceptable ("candidate") rules
**Output:** A chosen rule, to be used for aggregating the evaluations
1. Split the evaluators uniformly at random into two groups, considering each group as a copy of the process (in line with the peer review experiments above);
2. For each candidate rule, compute the rule's outputs separately on the two groups, and measure the disagreement between these two outputs; return the rule with min. disagreement

---

Importantly, Algorithm 1 is agnostic to the types of input/output of the rules it is picking among; it only requires a measure of disagreement for their outputs (Step 2). It is therefore widely applicable in settings with various evaluation formats—*e.g.* rankings, ratings/scores, approval sets—and desired outputs—*e.g.*, an aggregate ranking, a single winner, a reward function. In this paper, we focus on rules that output an aggregate ranking, providing a concrete analysis and experiments.

As we will show, $AbC$ has many advantages. First, it satisfies important axioms for RPRs, even those failed by many natural RPRs. Second, $AbC$ does *not* assume a ground truth; however, if a generative model indeed approximates the data, as we show experimentally, its MLE yields high consistency across random splits. $AbC$ then chooses this MLE, obtaining the benefits of the statistical approach. Third, many social choice axioms can be *imposed* on $AbC$ by restricting the candidate rules to those satisfying them, obtaining the benefits of the axiomatic approach. Lastly, any $AbC$ implementation is easy to continually extend by implementing new rules and adding them to our candidate rules.

## 3 PRELIMINARIES AND RULE PICKING RULES

We consider a set of *voters* $N = [n]$ and a set of *alternatives* $A$ with $|A| = m$. For our axiomatic analysis (Section 5), it will be helpful to assume each voter ranks *all* alternatives, although our method easily extends to other evaluations formats, including partial rankings (Section 4) and ratings (Section 7). A *weak ranking* is a complete and transitive binary relationship on $A$. A *strict ranking* is a weak ranking that is also asymmetric. Intuitively, a weak ranking is a strict ranking where ties are allowed. We denote the set of all strict (resp. weak) rankings of $A$ by $\mathcal{L}(A)$ (resp. $\mathcal{R}(A)$). For a (strict/weak) ranking $r$, we write $a \succ_r b$ if $r$ ranks $a$ strictly above $b$, and $b \succeq_r a$ otherwise. Each voter $i \in N$ has a strict ranking $\sigma_i \in \mathcal{L}(A)$. A *profile* $\boldsymbol{\sigma} \in \mathcal{L}(A)^n$ contains the rankings of all voters.

**Candidate rules** A *social welfare function (SWF)* is a mapping $f$ that, given a profile $\boldsymbol{\sigma}$, outputs a single weak ranking.[2] To aggregate voters' rankings, we are interested in picking an SWF to use from a set of acceptable "candidate" rules (*e.g.*, those that are the MLE of a noise model, or voting rules that satisfy certain axioms). From a machine-learning perspective, the candidate rules can be viewed as our hypothesis class. Importantly, our framework does not place any restrictions on the candidate rule set. We next introduce one such class of SWFs we will sometimes pay attention to.

**Definition 1.** A (monotonic) *positional scoring rule* is an SWF $f_s$ associated to a vector $s = (s_i)_{i \in [m]}$ with $1 = s_1 \geq \ldots \geq s_m = 0$. Given a profile $\boldsymbol{\sigma}$, for each alternative $a \in A$ and $i \in [m]$, say $M_{\boldsymbol{\sigma}}[a, i]$ is the number of voters ranking $a$ in the $i^{\text{th}}$ position, and $t_{\boldsymbol{\sigma}}^s[a] = \sum_{i=1}^m s_i M_{\boldsymbol{\sigma}}[a, i]$. Then, $f_s$ ranks $a \succ_{f_s(\boldsymbol{\sigma})} b$ iff $t_{\boldsymbol{\sigma}}^s[a] > t_{\boldsymbol{\sigma}}^s[b]$. We denote the set of all positional scoring rules by $F_S$.

Positional scoring rules are easy to represent/compute, and include well-known SWFs, *e.g.*, *plurality* ($f_p$, for $p = (1, 0, \ldots, 0)$), *veto* ($f_v$, for $v = (1, \ldots, 1, 0)$), and *Borda count* ($f_b$, for $b_i = \frac{m-i}{m-1}$).

**Rule Picking Rules** We now introduce a novel framework for picking rules.

**Definition 2.** A *rule picking rule (RPR)* is a function $Z$ that given a set of SWFs $F$ (called *candidate rules*) and a profile $\boldsymbol{\sigma}$, outputs a subset of rules $Z(F, \boldsymbol{\sigma}) \subseteq F$.

Just as many natural SWFs (*e.g.*, positional scoring rules) can lead to ties among alternatives, as we will see, many natural RPRs may tie some SWFs for certain profiles, leading to $|Z(F, \boldsymbol{\sigma})| > 1$. In such cases, a tie-breaking order over $F$ can be used to pick the single rule to be adopted.

RPRs offer a principled way to pick an SWF for aggregating the rankings in $\boldsymbol{\sigma}$. Ideally, an RPR will capture the idea that different (perfectly reasonable) SWFs may be appropriate for different profiles, *e.g.*, because the profile resembles a specific distribution, or certain positions in the rankings are more informative than others. Section 5 formalizes these cases by defining natural axioms for RPRs.

## 4 AGGREGATION BY CONSISTENCY (*AbC*)

Having introduced our framework of RPRs, we now formally present our method. As discussed in Section 2, we want our RPR to maximize the consistency between two independent sets of rankings. Our approach can be implemented with any measure of distance comparing the outputs of a rule. For concreteness in our analysis and experiments, we turn to one such well-known measure.

**Definition 3.** For weak rankings $r_1, r_2 \in \mathcal{R}(A)$ and alternatives $a, b \in A$, let $D_{r_1, r_2}^{a,b}$ be indicator variable that $r_1$ and $r_2$ strictly disagree about how $a$ and $b$ should be ordered. Similarly, let $T_{r_1, r_2}^{a,b}$ indicate that $a$ and $b$ are tied by at least one of $r_1$ or $r_2$. More formally, $D_{r_1, r_2}^{a,b} = \mathbb{I}[(a \succ_{r_1} b \text{ and } b \succ_{r_2} a) \text{ or } (b \succ_{r_1} a \text{ and } a \succ_{r_2} b)]$ and $T_{r_1, r_2}^{a,b} = \mathbb{I}[(a \succeq_{r_1} b \text{ and } b \succeq_{r_1} a) \text{ or } (a \succeq_{r_2} b \text{ and } b \succeq_{r_2} a)]$. The *Kendall tau distance with ties* between $r_1$ & $r_2$ is $KT(r_1, r_2) = \sum_{\{a,b\} \in A^2 : a \neq b} \left( D_{r_1, r_2}^{a,b} + \frac{1}{2} T_{r_1, r_2}^{a,b} \right)$.

$KT$ is more traditionally defined for strict rankings, without the $T_{r_1, r_2}^{a,b}$ term. We explicitly add the term for ties in order to penalize indecisiveness (*e.g.*, a rule that always returns all alternatives tied). Weighing ties by $\frac{1}{2}$ is inspired by Kendall's Tau-b correlation coefficient (Kendall, 1945). To study the impact of ties, we rerun some of our experiments with different weights for ties in Appendix C.8.

We are now ready to introduce our consistency-based RPR, which we formally define in Box 1.

---

[2]Outputting weak rankings allows returning a single ranking without violating neutrality or anonymity.

---

**Box 1: Aggregation by Consistency ($AbC$)**

Given a profile $\boldsymbol{\sigma}$, consider the following random process: Initialize two sets $N_1 = N_2 = \emptyset$. For each voter $i \in N$, pick $j \in \{1, 2\}$ uniformly at random and set $N_j \leftarrow N_j \cup \{i\}$. Let $\boldsymbol{\sigma}^{(j)} = \{\sigma_i\}_{i \in N_j}$ (*i.e.*, the restriction of profile $\boldsymbol{\sigma}$ to voters in $N_j$) for each $j \in \{1, 2\}$. Then, given a set of candidate rules $F$, *Agreement by Consistency (AbC)* is an RPR defined as

$$AbC(F, \boldsymbol{\sigma}) = \underset{f \in F}{\arg\min} \, \mathbb{E}\left[ KT\left( f(\boldsymbol{\sigma}^{(1)}), f(\boldsymbol{\sigma}^{(2)}) \right) \right], \tag{1}$$

where the expectation is over splitting $\boldsymbol{\sigma}$ into $\boldsymbol{\sigma}^{(1)}$ and $\boldsymbol{\sigma}^{(2)}$ by process described above.

---

In words, $AbC$ returns the SWFs among $F$ that minimize, in expectation,[3] the disagreement when applied separately to two sides of a random split of $\boldsymbol{\sigma}$. The next example illustrates this.

**Example 4.** *Fix some integer $k \geq 2$ and consider the profile $\boldsymbol{\sigma}$ with alternatives $A = \{a, b, c\}$ and $n = 3k$ voters. The voters comprise three groups of size $k$ each, with the following preferences:*
 • *Group 1 ranks $a \succ b \succ c$.* • *Group 2 ranks $a \succ c \succ b$.* • *Group 3 ranks $b \succ c \succ a$.*
*Consider the set of candidate rules $F = \{f_p, f_v\}$ (i.e., plurality and veto). Say $\boldsymbol{\sigma}^{(1)}$ and $\boldsymbol{\sigma}^{(2)}$ are the random variables resulting from splitting $\boldsymbol{\sigma}$ via the random process in Box 1. First, consider $f_p$: say $r_p^1 = f_p(\boldsymbol{\sigma}^{(1)})$ and $r_p^2 = f_p(\boldsymbol{\sigma}^{(2)})$. Since alternative $c$ is not ranked top by any voter, both $r_p^1$ and $r_p^2$ will rank $a \succ c$ (resp. $b \succ c$) unless all of the $2k$ (resp. $k$) voters ranking $a$ (resp. $b$) as their top alternative end up on the same side of split, in which case we get a tie. Thus,*

$$\mathbb{E}\left[D_{r_p^1, r_p^2}^{a,c}\right] = \mathbb{E}\left[D_{r_p^1, r_p^2}^{b,c}\right] = 0; \quad \mathbb{E}\left[T_{r_p^1, r_p^2}^{a,c}\right] \leq \frac{1}{2^{2k-1}}; \quad \mathbb{E}\left[T_{r_p^1, r_p^2}^{b,c}\right] \leq \frac{1}{2^{k-1}}.$$

*Using the linearity of expectation, this implies the expectation in (1) for $f_p$ is $\mathbb{E}\left[KT\left(r_p^1, r_p^2\right)\right] =$*

$$\mathbb{E}\left[D_{r_p^1, r_p^2}^{a,b} + T_{r_p^1, r_p^2}^{a,b}/2 + D_{r_p^1, r_p^2}^{a,c} + T_{r_p^1, r_p^2}^{a,c}/2 + D_{r_p^1, r_p^2}^{b,c} + T_{r_p^1, r_p^2}^{b,c}/2\right] \leq 1 + \frac{1}{2^{2k}} + \frac{1}{2^k} \leq 1.32. \tag{2}$$

*Now, consider $f_v$: say $r_v^1 = f_v(\boldsymbol{\sigma}^{(1)})$ and $r_v^2 = f_v(\boldsymbol{\sigma}^{(2)})$. Since the number of voters that rank $a$ and $b$ bottom are tied, $r_v^1$ and $r_v^2$ will either disagree about $a$ and $b$, or have them tied. Thus, we cannot have $D_{r_v^1, r_v^2}^{a,b} = T_{r_v^1, r_v^2}^{a,b} = 0$. The same is true for any other pair of alternatives. This implies*

$$\mathbb{E}\left[KT\left(r_v^1, r_v^2\right)\right] = \mathbb{E}\left[D_{r_v^1, r_v^2}^{a,b} + T_{r_v^1, r_v^2}^{a,b}/2 + D_{r_v^1, r_v^2}^{a,c} + T_{r_v^1, r_v^2}^{a,c}/2 + D_{r_v^1, r_v^2}^{b,c} + T_{r_v^1, r_v^2}^{b,c}/2\right] \geq 1.5. \tag{3}$$

*Comparing (2) and (3), we see that $AbC(F, \boldsymbol{\sigma}) = \{f_p\}$. Choosing plurality over veto indeed makes sense for $\boldsymbol{\sigma}$, as the former clearly gives more information on how alternatives compare to each other.*

We now discuss three **extensions** of $AbC$, which we later implement (Section 7). As computing the expected disagreement in (1) for each SWF can be difficult for more complicated profiles and rules than in Example 4, in practice $AbC$ can be approximated for finite $|F|$ via **Monte Carlo sampling**, *i.e.*, by splitting voters via the random process in Box 1, computing $KT(f(\boldsymbol{\sigma}^{(1)}), f(\boldsymbol{\sigma}^{(2)}))$ for each $f \in F$, and repeating for a desired number of splits, eventually returning the SWF with the minimum average disagreement. **Even for infinite** $|F|$, optimization/learning algorithms can be used to find the minimizer of (1). For example, if $F = F_S$ (Def. 1), one can run a constrained optimization with the scoring vector $(s_i)_{i \in [m]}$ as variables. Indeed, our $AbC$ implementation in Section 7 uses Monte Carlo sampling and includes (among other rules) an optimization over all positional scoring rules.

A more general setting is that of **partial rankings**, where each voter $i \in N$ ranks a subset $A_i \subseteq A$. This setup is more appropriate for certain settings where we would like to utilize $AbC$, such as peer review and RLHF. To extend $AbC$ to partial rankings, we must take into account that each alternative is no longer ranked by every voter. Thus, some alternatives may have more evaluations on one side of a random split, while others' evaluations will be more evenly split. Ideally, we should penalize disagreements over the former alternatives less harshly, as one side of the split may lack sufficient information about them. Indeed, in the extreme case where all evaluations of an alternative are on the same side, it is unreasonable to expect any SWF to rank this alternative consistently across the split. Thus in Section 7, we replace $KT$ in (1) with the *weighted KT function* (Kumar & Vassilvitskii, 2010), setting the weight of an alternative according to how evenly it is represented across a split.

---

[3] In Box 1, there is a $2^{-(n-1)}$ probability that $N_i = \emptyset$ for some $i \in \{1, 2\}$. In this case, we take $f(\boldsymbol{\sigma}^{(i)})$ to be the empty ranking (all alternatives tied) for all $f \in F$. This effectively gives zero weight to all such splits.

# 5    AXIOMATIC ANALYSIS OF *AbC*

We next investigate the behavior and properties of *AbC* by introducing natural axioms for RPRs.

## 5.1    CONSISTENCY AXIOMS

As RPRs intend to identify rules that are appropriate to a given profile, it is natural to seek RPRs that behave consistently with *changes* to the profile. Below, we define axioms that formalize this.

**Reversal symmetry:** One basic change to a profile is to flip every voter's ranking. In this case, we would like the SWFs picked by a RPR to flip too. Formally, for a positional scoring rule $f_s \in F_S$, say $\mathrm{rev}(f_s)$ is the scoring rule $f_{s'}$ associated with $s' = (1 - s_m, \ldots, 1 - s_1)$, *e.g.*, the reverse of plurality is veto, whereas Borda count is its own reverse. For $F \subseteq F_S$, we write $\mathrm{rev}(F) = \{\mathrm{rev}(f) : f \in F\}$. For any ranking $r \in \mathcal{L}(A) \cup \mathcal{R}(A)$, say $\mathrm{rev}(r)$ is the reversed ranking (*e.g.*, $a \succ b \succ c$ becomes $c \succ b \succ a$). Last, say $\mathrm{rev}(\boldsymbol{\sigma}) = \{\mathrm{rev}(\sigma_i)\}_{i \in N}$ for any profile $\boldsymbol{\sigma}$. We now define reversal symmetry, inspired by the homonymous property for rank aggregation methods by Saari (1994).

**Definition 5.** An RPR $Z$ satisfies *reversal symmetry* if for any subset of positional scoring rules $F \subseteq F_S$ such that $\mathrm{rev}(F) = F$ and for all profiles $\boldsymbol{\sigma}$, we have $\mathrm{rev}(Z(F, \boldsymbol{\sigma})) = Z(F, \mathrm{rev}(\boldsymbol{\sigma}))$.

To see why this is a natural property, suppose the "signal" in the votes is concentrated at the top: for each voter, only the choice of the top alternative is statistically informative. In that case, a reasonable RPR should choose plurality. But if we flip all the votes, the signal is concentrated at the bottom, and a reasonable RPR should choose veto. As shown in Theorem 1 below, *AbC* satisfies this axiom.

**Shuffling consistency:** As with the example above, in some profiles certain parts of the rankings may be more informative than others. For example, if we uniformly "shuffle" an interval of positions in every voter's ranking, exactly where in this interval an alternative ends up reveals no information on how it compares to other alternatives in the interval. Thus, for such a "shuffled profile," we would want our RPR to pick rules that treat these positions equivalently. We now formally define shuffling.

**Definition 6.** Given a profile $\boldsymbol{\sigma}$ and a subset $S \subseteq \{1, 2, \ldots, m\}$, the $k$-*shuffling of $\boldsymbol{\sigma}$ with respect to* $S$ (denoted $\pi^k(\boldsymbol{\sigma}, S)$) is a profile obtained as follows: For each voter $i \in N$, (1) Create $k \cdot m!$ copies of voter $i$'s ranking $\sigma_i$ and separate them into $|S|!$ groups of equal size; (2) Assign each group to a unique permutation of $S$. Modify the votes in each group so that the candidates in the positions in $S$ are permuted according to the assigned permutation; (3) Add all copies to the final profile.

For instance, if $\boldsymbol{\sigma}$ consists of only $a \succ b \succ c$, then $\pi^1(\boldsymbol{\sigma}, \{1, 2\})$ has six rankings, three $a \succ b \succ c$ and three $b \succ a \succ c$. Our next axiom looks at the extreme case of shuffling all but a single position.

**Definition 7.** Say we are given a finite set of positional scoring rules $F \subseteq F_S$ containing plurality (*i.e.*, $f_p \in F$) and a profile $\boldsymbol{\sigma}$ such that $f_p(\boldsymbol{\sigma})$ has no ties. Then, an RPR $Z$ satisfies *plurality-shuffling consistency (PSC)* if for any such $F$ and $\boldsymbol{\sigma}$, there is a $k \in \mathbb{Z}_+$ so that $Z(F, \pi^{k'}(\boldsymbol{\sigma}, [2, m])) = \{f_p\}$ for all $k' \geq k$, *i.e.*, $Z$ picks *only* $f_p$ for the shuffling with respect to $[2, m] = \{2, 3, \ldots, m\}$.

Intuitively, for a (sufficiently large) profile where the top positions of the votes gives an unambiguous ranking, but the remaining $m - 1$ positions are effectively indistinguishable, an RPR satisfying PSC identifies the rule that treats these $m - 1$ positions equally (plurality) as the only appropriate rule. Such a profile can approximate settings where the voters exclusively know/state their top alternative, *e.g.*, in certain types of RLHF queries (Xu et al., 2024). PSC is a natural property that respects the symmetry in the profile and identifies where the "signal" in the votes is concentrated. Despite this, a large class of RPRs, which we introduce next, all fail PSC, as we will show in Theorem 1.

**Definition 8.** An RPR $Z$ is *welfare-maximizing* if there exists $u : \mathcal{L}(A) \times \mathcal{R}(A) \to \mathbb{R}$ such that $Z(F, \boldsymbol{\sigma}) = \arg\max_{f \in F} u(\boldsymbol{\sigma}, f(\boldsymbol{\sigma}))$ for all $\boldsymbol{\sigma}$ and $F$, where we write $u(\boldsymbol{\sigma}, r) = \sum_{i=1}^n u(\sigma_i, r)$.

Welfare-maximizing RPRs capture a common approach: interpreting the optimal voting rule as one that maximizes social welfare with respect to some utility function (Caragiannis & Procaccia, 2011; Gershkov et al., 2017). As Definition 8 puts no restrictions on the function $u$, welfare-maximizing RPRs constitute a wide class. Nevertheless, we will show *all* welfare-maximizing RPRs fail PSC.

On the other hand, this is not the case for *AbC*. To see this, take a random split of the shuffled profile $\pi^k(\boldsymbol{\sigma}, [2, m])$. Given the balanced nature of all positions except the first, any rule that treats a dis-

tinction between alternatives in these positions as a signal will inevitably observe the reversed signal on the other side of the split, leading to a higher disagreement. We formalize this in Theorem 1.

**Union consistency:** Next, we study the behavior of RPRs when we combine sets of voters, inspired by an analogous axiom for voting rules, simply named consistency (Young, 1975). Informally, it dictates that whenever an alternative is the winner of a voting rule applied to the rankings of two distinct sets of voters, the same alternative should still win when we bring those sets of voters together. We now define our version of the axiom, named *union consistency*, specifically for RPRs.

**Definition 9.** For two profiles $\boldsymbol{\sigma}_a$ and $\boldsymbol{\sigma}_b$ over the same alternatives but with two disjoint sets of voters $N_a$ and $N_b$, say $\boldsymbol{\sigma}_a + \boldsymbol{\sigma}_b$ is the profile with all voters $N_a \sqcup N_b$. An RPR $Z$ satisfies *union consistency (UC)* if $Z(F, \boldsymbol{\sigma}_a) \cap Z(F, \boldsymbol{\sigma}_b) \neq \emptyset$ implies $Z(F, \boldsymbol{\sigma}_a + \boldsymbol{\sigma}_b) = Z(F, \boldsymbol{\sigma}_a) \cap Z(F, \boldsymbol{\sigma}_b)$.

Unlike previous axioms, $AbC$ does not satisfy UC, but (as we show below) neither does any RPR satisfying reversal symmetry and PSC, giving us our first impossibility result for RPRs. Further, we argue that union consistency is less significant for an RPR than its counterpart for voting rules: just because a rule is appropriate for two different sets of rankings does not necessarily mean it is appropriate for their union, especially if they are differently distributed. Indeed, in the extreme case of a single voter, *any* sensible aggregation rule (*i.e.* that returns the ranking itself) is appropriate. We now present Theorem 1, which states our results regarding the axioms in this subsection.

**Theorem 1.** *(I) $AbC$ satisfies reversal symmetry. (II) Any welfare-maximizing RPR fails PSC; $AbC$ satisfies it. (III) No (anonymous) RPR can satisfy all three of reversal symmetry, PSC, and UC.*

*Proof sketch.* The proof can be found in Appendix B.1. (I) follows from the symmetries of the $KT$ function. (II) is proven by showing plurality cannot achieve better welfare than any other positional scoring rule in a shuffled profile. For (III), we show that the difference between the total scores assigned to two alternatives by any fixed rule on the shuffled profile is the weighted sum of a several (shifted) binomials. A careful application of tight lower and upper tail bounds for binomials (Ash, 1990) then shows that the disagreement of plurality is strictly lower than that of any other rule. $\square$

## 5.2 PRESERVED AXIOMS

While using an RPR has advantages beyond just outputting an aggregate ranking (see Section 1), it is true that an RPR $Z$, when paired with a set of candidate rules $F$, *induces* an SWF: $f_Z^F(\boldsymbol{\sigma}) \overset{\text{def}}{=} f(\boldsymbol{\sigma})$ where $Z(F, \boldsymbol{\sigma}) = \{f\}$. In words, $f_Z^F$ is the SWF that first maps a profile to an SWF using $Z$, and then applies that SWF to the profile. For $f_Z^F$ to be well defined, we must have $|Z(F, \boldsymbol{\sigma})| = 1$. As we are interested in the properties of $f_{AbC}^F$, for this subsection alone, we restrict our setting to profiles for which $|AbC(F, \boldsymbol{\sigma})| = 1$, and to RPRs $Z$ for which $|Z(F, \boldsymbol{\sigma})| = 1$ for these profiles.

Clearly, the axiomatic properties of $f_Z^F$ will depend on our choice of the candidate rules $F$. By selecting $F$ that all satisfy a certain property, can we ensure that so will $f_Z^F$? More formally, we say an RPR $Z$ *preserves* a property $P$ if whenever $P$ is true for all $f \in F$, then $P$ is true for $f_Z^F$. For example, recall that an SWF is neutral (resp. anonymous) if permuting the alternatives $A$ (resp. the voters $N$) in a profile results in the output ranking being permuted in the same way (resp. the output ranking not changing). It is immediate from Box 1 that $AbC$ preserves these two properties. On the other hand, this is not true for all RPRs, *e.g.*, for any $a \in A$, say $Z_a$ maps each $\boldsymbol{\sigma}$ to the SWF $f \in F$ that ranks $a$ highest in $f(\boldsymbol{\sigma})$. Then, $f_{Z_a}^F$ is clearly not neutral, even if all $f \in F$ are. Still, as show in Theorem 2 below, certain fundamental social choice axioms are preserved by *all* RPRs.

Not all axioms are as easily preserved. An SWF satisfies *monotonicity* if promoting an alterative in a voter's ranking while keeping all else constant does not hurt that alternative in the output ranking (*cf.* Appendix B.2 for a formal definition). It turns out $AbC$ does *not* preserve monotonicity. This is because promoting an alternative may cause $AbC$ to now pick a different rule ranking that alternative further below, even if it does not hurt that alternative under any specific rule. Still, much like union consistency, we show that preserving monotonicity is incompatible with an axiom that $AbC$ satisfies.

**Theorem 2.** *(I) $AbC$ preserves anonymity & neutrality. (II) Any RPR preserves the Smith criterion, Condorcet consistency, majority winner, pairwise majority consistency, and unanimity. (III) No (anonymous) RPR can satisfy reversal symmetry and preserve monotonicity.*

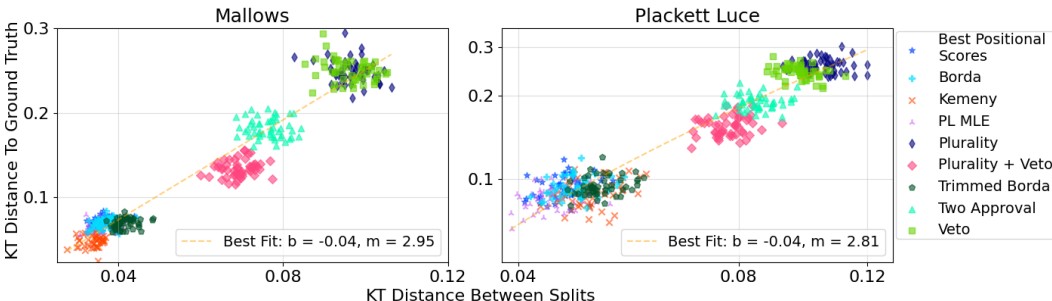

Figure 1: Log-log plots of $KT$ between ground truth and the ranking generated by SWFs on the complete profile (*i.e.*, error) vs. $KT$ between rankings generated by SWFs on splits of the data (*i.e.*, inconsistency) on partial rankings drawn from the Mallows & Plackett-Luce models. Each point shows average distance over 10 splits for a single profile. The plots display a clear positive relationship. In both cases, the MLE of the model outperforms other SWFs in both axes. As $AbC$ picks the rule with the min. distance between splits, it would pick the model's MLE in both cases.

For readers unfamiliar with the axioms in (II), we provide the formal definitions in Appendix B.2, along with the proof of the full theorem. (II) demonstrates a strength of not just $AbC$, but of our RPR framework, showing how one can still reap the benefits of the axiomatic approach by restricting the candidate rule set to certain SWFs. For proving (III), we construct an explicit profile in which promoting an alternative results in the reverse of the profile, and show that the demands of reversal symmetry and monotonicty cannot be simultaneously satisfied on this profile.

Overall, we have shown that $AbC$ satisfies many natural axioms (reversal symmetry, plurality-shuffling consistency, and preservation of the axioms in Theorem 2); as for axioms it fails (union consistency and the preservation of monotonicity), we have given impossibility results showing each of them is incompatible with axioms that $AbC$ satisfies. In Appendix C.9, we apply the data-driven approach of Caiata et al. (2025) to show experimentally that violations of monotonicity and union consistency are typically rare across well-known distributions, as well as on real-life data.

## 6 COMPUTATIONAL PROBLEM: PERFPOS

$AbC$ picks the SWF(s) from $F$ that minimizes the expected disagreement over a random split (Box 1). In this section, we show that when $F = F_S$ (positional scoring rules), the algorithmic problem of minimizing disagreement over even a given split is hard. Say we are given a ranking $\sigma_i \in \mathcal{L}(A)$ for each voter $i \in N$, as well as an even split of voters $N = N_1 \sqcup N_2$ with $|N_1| = |N_2|$. Then, PERFPOS (**Perf**ect **Pos**itional) asks: Is there a positional scoring rule $f_s \in F_S$ that achieves perfect consistency over this split, *i.e.*, obtains $KT(f_s(\boldsymbol{\sigma}^{(1)}), f_s(\boldsymbol{\sigma}^{(2)})) = 0$?

**Theorem 3.** PERFPOS *is* NP-*complete.*

*Proof sketch.* The proof, which is a reduction from 3SAT, can be found in Appendix B.3. Starting from a 3CNF formula $\phi$, we explicitly construct two half-profiles (corresponding to the two sides of a split) with pairs of alternatives for each clause and each binary variable in $\phi$. The positions of the pairs associated with the variables force the difference between successive components of the score vector to be either sufficiently small or sufficiently large, corresponding to the associated variable being assigned True or False. The pairs of alternatives associated with the clauses then ensures that perfect consistency can be reached if and only if this assignment of binary variables satisfies $\phi$. □

This shows that when $F = F_S$, the problem of computing the minimal possible disagreement is not only hard, but also hard to approximate to any multiplicative factor (since any algorithm with such a guarantee must return a solution with disagreement 0 when possible). While technically distinct, our result is aligned with other hardness results for optimizing over positional scoring rules, *e.g.*, for picking the rule most consistent with an underlying true ranking (Caragiannis et al., 2019).

Table 1: Mean and St. Dev. of $KT$ distance over 1000 splits from several metrics on review scores provided by Kerzendorf et al. (2020). While the max. score of items are often prioritized in peer review (Nierstrasz, 2000), $AbC$ shows that mean functions lead to more consistent outcomes here.

| Arithmetic Mean | Min | Max | Median | Geometric Mean |
|---|---|---|---|---|
| $0.364 \pm 0.001$ | $0.444 \pm 0.001$ | $0.409 \pm 0.001$ | $0.371 \pm 0.001$ | $0.369 \pm 0.001$ |

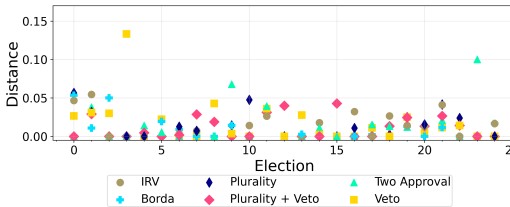

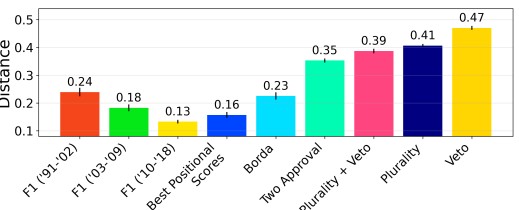

(a) Distance between splits of rankings in empirical election data from cities using Instant Runoff Voting (IRV). Details on data in this plot are found in Table 2. The consistency ordering of the rules differs from election to election, and IRV may perform worst. As $AbC$ picks the rule with min. disagreement for each election, it is able to reliably select a consistent ranking.

(b) Distance between splits for SWFs aggregating rankings of drivers in F1 races, where each race plays the role of voter. Rules labeled F1 correspond to scoring vectors used in practice and are evaluated for only the years in which that rule was in use. $AbC$ can be used to evaluate the impact of rule changes; in this case, both changes in F1 improved consistency.

Figure 2: Split distance with several rules on data from political elections and Formula One races.

## 7 EXPERIMENTS

Despite our complexity result, $AbC$ can be efficiently implemented using Monte Carlo sampling, even when *all* positional scoring rules are included in candidate rules. We now give experimental results evaluating $AbC$ and discuss their implications. See Appendix C for additional details regarding the experiments, including specific datasets, distributions, and parameter settings.

**Ground truth distance vs. disagreement:** While $AbC$ does *not* assume a ground truth, if a generative model is a reasonable approximation of the data, it performs desirably: Figure 1 highlights a clear positive relationship between distance to ground truth and disagreement between splits for several SWFs under partial rankings drawn from two well-known distributions, confirming our intuition from MVUEs (Section 2). The SWFs we test include an optimization for minimizing disagreement over all positional scoring rules ("Best Positional Scores", discussed further below). Importantly, for both Mallows and Plackett-Luce, the model's MLE (Kemeny and PL MLE, respectively) outperforms other SWFs in both axes. As each MLE has the lowest disagreement for its model, $AbC$ would pick it when given data from this model, hence obtaining the benefits of the statistical approach.

**Score data:** $AbC$ can also be applied when evaluators submit ratings/scores rather than rankings; we simply split the ratings received by each alternative via the random process in Box 1. Running $AbC$ then provides insights into functions from ratings to aggregate rankings. Table 1 shows the mean disagreement of score aggregation functions on the astronomy peer review dataset provided by Kerzendorf et al. (2020). While past work suggests items with a high maximum score are often prioritized in peer review (Nierstrasz, 2000), $AbC$ shows mean functions provide better consistency.

**Peer review of proposals:** Peer review data can also take the form of rankings. In Appendix C.2, we run $AbC$ on anonymized rankings provided by evaluators of projects proposed for the Atacama Large Millimeter Array (ALMA) (Meyer et al., 2022), which is the largest ground-based radio telescope on earth. Our results reveal that modifications previously considered by ALMA—*e.g.*, adding outlier rejection to Borda count— significantly hurt the consistency of their aggregation rule.

**Aggregating contests:** In some settings such as evaluating AI agents (Lanctot et al., 2025a), contestants compete in several events (each of which acts as a voter), and how to best aggregate the results into an overall ranking is unclear. Indeed, Formula One (F1) has changed the positional scoring rule used for aggregating races several times (Boehmer et al., 2022). With $AbC$, we measure the impact of such changes on consistency (Fig. 2b); an analogous experiment on Olympics is in Appendix C.4.

**Political elections:** Figure 2a considers running $AbC$ on several political elections, each with thousands of voters. Each election corresponds to a single value on the x-axis; for each election we show the $KT$ distance of each SWF (*e.g.*, veto, indicated by the yellow square, gives a distance of 0.14 on election 3). In each election, the real-world winner was determined using Instant Runoff Voting (IRV). However, in some of these elections (*e.g.*, 1,16,18) IRV is the rule that gives the *worst* disagreement. Importantly, in 21 out of the 25 elections, there is a rule that achieves zero disagreement; however, this rule is different between elections, confirming our intuition that different rules are appropriate for different settings (Section 1). As a result, $AbC$ reliably selects a consistent ranking.

**Recommender systems:** In order to demonstrate the wide applicability of $AbC$ across a diverse set of domains, Appendix C.6 reports the result of running our method on the MovieLens dataset (Harper & Konstan, 2015), which is used widely to evaluate recommender systems. As shown there, the most consistent rankings balance high (perceived) quality with wide appeal.

**Other distance functions:** While we have defined $AbC$ using the Kendall-Tau distance function for our axiomatic analysis in Section 5, the $AbC$ framework can also be used in conjunction with other distance functions, as mentioned in Section 4. The choice of the distance function can depend on the specific setting and the goals of the implementer. For example, if disagreements on top of the output ranking should be prioritized over disagreements in lower positions (*e.g.*, we are aggregating search results from different algorithms for a meta-search engine (Dwork et al., 2001)), then *Normalized Discounted Cumulative Gain* can be used to measure disagreement. If, on the other hand, the desired output is not an aggregate ranking but instead a subset of alternatives of a fixed size (*e.g.*, we are picking $k < m$ proposals to fund, and the ranking among the winners is not important), then *Jaccard dissimilarity* can be used as a distance measure between the chosen subsets. In Appendix C.7, we rerun the experiments for the ALMA and F1 data using these two alternative distances to show they produce qualitatively similar results. We believe studying the axiomatic properties of $AbC$ for different output formats and different distance functions is an important avenue of future research.

**Optimizing positional scoring rules:** While deciding whether a positional scoring rule leads to zero disagreement is NP-complete, we can, in practice, learn positional score vectors which approximately minimize the $KT$ distance for sampled splits. Figure 3 shows the $KT$ distance of the best vector found by two optimization methods on Formula One race data: Stochastic Gradient Descent (SGD) and Simulated Annealing. While both methods find score vectors which improve upon their initial state, we use Simulated Annealing in the experiments above as it typically finds a score vector with lower disagreement and requires much less compute time than SGD.

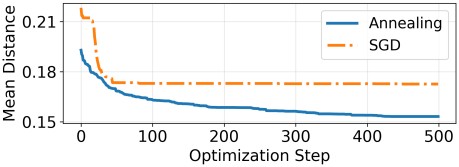

Figure 3: SGD and Simulated Annealing performance during optimization of KT distance for Formula One race data. Annealing outperforms SGD in terms of finding a rule with low disagreement / high consistency.

Overall, our method provides a principled way in which users can evaluate the impact of modifications to existing methods by running $AbC$ on their own data. In some cases, the answer is that it is an improvement: both changes to F1 rules significantly reduced disagreement (Figure 2b). In others, the proposed modification hurts consistency: Meyer et al. (2022) consider adding outlier rejection to Borda (removing the min. & max. scores of each alternative) for peer review, but our experiment suggests this method (Trimmed Borda) increases disagreement on their dataset (Appendix C.2).

## 8 CONCLUSION AND FUTURE WORK

Treating voters' rankings as fixed, we addressed the problem of picking the most appropriate rule. It is of interest to understand RPRs, and specifically $AbC$, more generally under strategic behavior of voters. Such behavior makes it harder to study the impact of a proposed rule change on consistency by running $AbC$ on prior data. Another direction of future work is to investigate RPRs under other axioms, either translated from axioms for rules or novel axioms for rule picking rules. Overall, RPRs (and $AbC$) open the door to principled, data-driven rule selection for diverse real-world applications.

ACKNOWLEDGEMENTS

We thank John Carpenter and Andrea Corvillon from the Atacama Large Millimeter/submillimeter Array (ALMA), as well as Kate Larson and Ariel Procaccia for helpful discussions. We also thank Sinan Karaböcüoğlu and Boğaçhan Arslan for implementing our method for the *2nd Computational Social Choice Competition* at the 33rd International Joint Conference on Artificial Intelligence (IJ-CAI 2024). R.E.B. and V.C. thank the Cooperative AI Foundation, Macroscopic Ventures (formerly Polaris Ventures / the Center for Emerging Risk Research), and Jaan Tallinn's donor-advised fund at Founders Pledge for financial support. R.E.B. is also supported by the Cooperative AI PhD Fellowship. The work of N.S. was supported by NSF CAREER award 1942124 and ONR N000142512346.lr

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

## A  ADDITIONAL RELATED WORK

In this section, we further elaborate on prior work conceptually related to our work.

**Rule selection**    The question of comparing different voting rules has been widely studied in the social choice literature. For an overview of the axiomatic and computational properties of various voting rules, see Brandt et al. (2016). There has been earlier empirical work on collecting voters' preferences over voting rules themselves (Sertel & Kara, 2003; Giritligil Kara & Sertel, 2005; Aldrich et al., 2014). However, it has been shown that (unsurprisingly) in many real-life cases voters simply prefer the rules that benefit their preferred candidate (Blais et al., 2015; Weber, 2020). In general, it is clear that even social choice theorists themselves cannot reach a consensus on what is the best voting rule (Laslier, 2012). There has also been interest in using tools from automated reasoning for instantiating and generating formal justifications for using a voting rule (Cailloux & Endriss, 2016) or imposing an axiom (Boixel & Endriss, 2020). Similar tools were used for bypassing voting rules altogether by picking a set of axioms that impose a certain outcome (Schmidtlein & Endriss, 2023).

**MLE approach**    The idea of treating votes as noisy estimates of a ground truth goes back to Condorcet [1785], who designed a noise model where every voter has a fixed probability of ranking each pair of alternatives correctly, and solved it for two and three alternatives. Young (1995) later extended this model to an arbitrary number of alternatives, showing that its MLE is equivalent to an earlier rule introduced by Kemeny (1959). Conitzer & Sandholm (2005) later studied the question of which voting rules are MLEs for some noise model where votes are sampled independently, in particular showing that any rule that violates consistency cannot be an MLE. Combined with the result that a continuous and neutral social choice function (rules outputting a set of winners) is consistent if and only if it is a scoring rule (Young, 1975), this shows that no other (neutral and continuous) social choice function can be the MLE of any noise model. Conitzer et al. (2009) did a similar analysis for social preference functions (rules outputting a set of rankings) providing an exact characterization of MLE rules as simple ranking scoring functions. Similar analyses followed for voting in multi-issue domains (Xia et al., 2010), with partial rankings (Xia & Conitzer, 2011), and on social networks (where voters no longer vote independently) Conitzer (2013). In the opposite direction, Azari Soufiani et al. (2014) study the MLE of two known noise models to identify whether they satisfy certain axiomatic properties, and Xia (2016) extend their results. Tideman & Plassmann (2014), on the other hand, construct a spatial noise model specifically to approximate data from actual elections, and evaluate the performance of common voting rules under this model. Lastly, a related but distinct approach to classify voting rules is distance-rationalizatability (DR) (Meskanen & Nurmi, 2008; Elkind et al., 2009), which requires a rule to map each election to the result of the closest consensus according to some metric. Elkind et al. (2010) combine the MLE and DR approaches to better understand and compare rules, and in this framework declare Kemeny to be the best rule, as it fits both frameworks with the same underlying function. It is worth noting, however, that DR only applies to social choice functions.

**Statistical model dependence in estimation from pairwise comparisons**    Another line of literature on model (in)dependence focuses on estimation from pairwise comparisons. Commonly used statistical models in this setting include the Bradley–Terry–Luce (BTL) (Bradley & Terry, 1952; Plackett, 1975) and Thurstone models (Thurstone, 1927), which all fall under the class of parameter-based models (also known as random utility models). However, recent studies (Shah et al., 2017; Shah & Wainwright, 2018; Heckel et al., 2019) have demonstrated that estimators derived from more general "permutation-based" models offer two notable advantages. First, when data are generated from these broader permutation-based models, these estimators exhibit significantly better performance. Second, even when data originate from parameter-based models such as BTL or Thurstone, the guarantees provided by these general estimators are within logarithmic factors of those achieved by estimators specifically tailored to parameter-based models.

## B  OMITTED PROOFS

In this section, we give the proofs that were omitted in the main body of the paper.

### B.1 PROOF OF THEOREM 1

We first recall our main result from Section 5.1, regarding the consistency axioms for RPRs.

**Theorem 1.** *(I) AbC satisfies reversal symmetry. (II) Any welfare-maximizing RPR fails PSC; AbC satisfies it. (III) No (anonymous) RPR can satisfy all three of reversal symmetry, PSC, and UC.*

We prove each claim in the theorem as a seperate proposition.

**Proposition 10.** *AbC satisfies reversal symmetry.*

*Proof.* Given any positional scoring rule $f_s \in F$ and a profile $\boldsymbol{\sigma}$, say $f_{s'}$ is the reverse rule of $f_s$, *i.e.*, $s' = (s'_1, s'_2, \ldots, s'_m) = (1 - s_m, 1 - s_{m-1}, \ldots, 1 - s_1)$. The total scores assigned to any alternative $a \in A$ by $f_{s'}$ when run of $\text{rev}(\boldsymbol{\sigma})$ (see Definition 1) is

$$t_{\text{rev}(\boldsymbol{\sigma})}^{s'}[a] = \sum_{j=1}^{m} s'_j M_{\text{rev}(\boldsymbol{\sigma})}[a, j] = \sum_{j=1}^{m} (1 - s_{m+1-j}) M_{\boldsymbol{\sigma}}[a, m+1-j]$$

$$= \sum_{j=1}^{m} (1 - s_j) M_{\boldsymbol{\sigma}}[a, j] = n - t_{\boldsymbol{\sigma}}^{s}[a],$$

implying that $f_s(\boldsymbol{\sigma}) = \text{rev}(f_{s'}(\text{rev}(\boldsymbol{\sigma}))) = \text{rev}(\text{rev}(f_s)(\text{rev}(\boldsymbol{\sigma})))$. Additionally, we note that the Kendall-Tau distance (Definition 3) is symmetric with respect to reversals, *i.e.*, $KT(r_1, r_2) = KT(\text{rev}(r_1), \text{rev}(r_2))$. Thus,

$$KT(f_s(\boldsymbol{\sigma}^{(1)}), f_s(\boldsymbol{\sigma}^{(2)})) = KT(\text{rev}(\text{rev}(f_s)(\text{rev}(\boldsymbol{\sigma}^{(1)}))), \text{rev}(\text{rev}(f_s)(\text{rev}(\boldsymbol{\sigma}^{(2)}))))$$

$$= KT(\text{rev}(f_s)(\text{rev}(\boldsymbol{\sigma}^{(1)})), \text{rev}(f_s)(\text{rev}(\boldsymbol{\sigma}^{(2)}))).$$

This implies that if $f_s$ minimizes equation 1 in Box 1 over random splits of $\boldsymbol{\sigma}$, then $\text{rev}(f_s)$ minimizes it over random splits of $\text{rev}(\boldsymbol{\sigma})$, and therefore $AbC(F, \boldsymbol{\sigma}) = \text{rev}(AbC(F, \text{rev}(\boldsymbol{\sigma})))$. $\square$

**Proposition 11.** *Any welfare-maximizing RPR $Z$ fails plurality-shuffling consistency.*

*Proof.* Take any $f_s \in F_S$. We will show that for any $k \in \mathbb{Z}^+$, we have $f_s(\pi^k(\boldsymbol{\sigma}, [2, m])) = f_p(\boldsymbol{\sigma})$, where $f_p$ is plurality. Recall from Definition 1 we have $s_1 = 1$ and $s_m = 0$ WLOG. By Definitions 1 and 6, the total score assigned to a given alternative $a \in A$ by $f_s$ on input $\pi^k(\boldsymbol{\sigma}, [2, m])$ (say $t_k[a]$) is

$$t_k[a] = M_{\boldsymbol{\sigma}}[a, 1] \cdot k \cdot m! s_1 + \sum_{i=2}^{m} M_{\boldsymbol{\sigma}}[a, i] \cdot \sum_{j=2}^{m} \frac{k \cdot m!}{(m-1)} s_j$$

$$= M_{\boldsymbol{\sigma}}[a, 1] \cdot k \cdot m! + \frac{k \cdot m!}{(m-1)} \left( \sum_{j=2}^{m-1} s_j \right) (n - M_{\boldsymbol{\sigma}}[a, 1])$$

$$= \frac{k \cdot m!}{(m-1)} \left( \sum_{j=2}^{m-1} s_j \right) n + M_{\boldsymbol{\sigma}}[a, 1] \cdot k \cdot m! \left( 1 - \frac{\sum_{j=2}^{m-1} s_j}{m-1} \right)$$

$$\equiv C + D \cdot M_{\boldsymbol{\sigma}}[a, 1],$$

where $C$ and $D$ does not depend on $a$. Since $1 = s_1 \geq s_2 \geq \ldots \geq s_m = 0$, we have $\sum_{j=2}^{m-1} s_j < m - 1$ and therefore $D > 0$. This implies that for any $a, b \in A$ we have

$$a \succ_{f_s(\pi^k(\boldsymbol{\sigma}, [m-1]))} b \Leftrightarrow t_k[a] > t_k[b]$$

$$\Leftrightarrow M_{\boldsymbol{\sigma}}[a, 1] > M_{\boldsymbol{\sigma}}[b, 1]$$

$$\Leftrightarrow a \succ_{f_p(\boldsymbol{\sigma})} b.$$

Since $f_s$ was arbitrarily chosen, this implies all rules in $F_S$ return the same output on $\pi^k(\boldsymbol{\sigma}, [m-1])$. Hence, for any welfare-maximizing RPR $Z$ and any $F \subseteq F_S$, we have $Z(F, \pi^k(\boldsymbol{\sigma}, [m-1])) = F$, showing that the rule fails Definition 7. $\square$

Since all $f_s \in F_S$ produce the same output on $\pi^k(\boldsymbol{\sigma}, [m-1])$, one might wonder whether it matters which rule an RPR returns. However, in practice an RPR might be employed to "lock in" a rule for future use, allowing us to use it in future profiles coming from the same source without re-running the RPR at each round, potentially increasing interpretability. Hence, with a plurality-shuffling-consistent rule, we would be able to keep on using $f_p$ in future profiles that are approximately (but not exactly) shuffled, as future repetitions of the same process are likely to once again have their signal concentrated in the top position. Moreover, identifying that no $f_s \in F_S$ provides any information beyond $f_p$ has a computational advantage: while the output of $f_p$ can be computed in $O(n+m)$ time, an arbitrary $f_s \in F_S$ might require $O(mn)$ time.

**Proposition 12.** *AbC satisfies plurality-shuffling consistency.*

*Proof.* Fix some finite $F \subset F_S$. If $|F| = 1$ we are done. Otherwise, given any profile $\boldsymbol{\sigma}$ (that satisfies the condition in Definition 7, *i.e.*, $f_p(\boldsymbol{\sigma})$ contains no ties) with $n = n$ voters an $m = m \geq 3$ alternatives,[4] define $\boldsymbol{\sigma}_k = \pi^k(\boldsymbol{\sigma}, [2, m])$ for all $k$. Let $M$ be the matrix such that for each alternative $a \in A$ and index $i \in [m]$, $M[a, i]$ indicates the number of voters in $\boldsymbol{\sigma}$ that rank $a$ in the $i^{\text{th}}$ position, and let $M_k$ be the analogous matrix for $\boldsymbol{\sigma}_k$.

For any $f \in F$ and any two candidates $a, b \in A$, say $D_{a,b}^{f,k}$ is the indicator random variable that the two output rankings resulting from applying $f$ to the two sides of a random split of $\boldsymbol{\sigma}_k$ into $\boldsymbol{\sigma}_k^{(1)}$ and $\boldsymbol{\sigma}_k^{(2)}$ (where each voter is independently and uniformly placed in one of the two sets, as defined in Box 1) disagree about $a$ and $b$'s position; *i.e.*,

$$D_{a,b}^{f,k} = \mathbb{I}[(a \succ_{f(\boldsymbol{\sigma}_k^{(1)})} b \text{ and } b \succ_{f(\boldsymbol{\sigma}_k^{(2)})} a) \text{ or } (b \succ_{f(\boldsymbol{\sigma}_k^{(1)})} a \text{ and } a \succ_{f(\boldsymbol{\sigma}_k^{(2)})} b)].$$

Similarly, say $T_{a,b}^{f,k}$ is the indicator random variable that $f$ ties $a$ and $b$ on one of the two profiles; *i.e.*,

$$T_{r_1,r_2}^{a,b} = \mathbb{I}[(a \succeq_{f(\boldsymbol{\sigma}_k^{(1)})} b \text{ and } b \succeq_{f(\boldsymbol{\sigma}_k^{(1)})} a) \text{ or } (a \succeq_{f(\boldsymbol{\sigma}_k^{(2)})} b \text{ and } b \succeq_{f(\boldsymbol{\sigma}_k^{(2)})} a)].$$

Then, using linearity of expectation and the fundamental bridge, we get

$$\mathbb{E}\left[KT(f(\boldsymbol{\sigma}_k^{(1)}), f(\boldsymbol{\sigma}_k^{(2)}))\right] = \mathbb{E}\left[\sum_{a,b \in A} D_{a,b}^{f,k} + \frac{1}{2} T_{a,b}^{f,k}\right] = \sum_{a,b \in A} \Pr\left[D_{a,b}^{f,k} = 1\right] + \frac{\Pr\left[T_{a,b}^{f,k} = 1\right]}{2}. \tag{4}$$

The rest of the proof will follow from the next lemma.

**Lemma 13.** *For any $f \in F \setminus \{f_p\}$ there exists a $k_f \in \mathbb{Z}^+$ such that*

1. *for all $k \geq k_f$ we have $\Pr\left[D_{a,b}^{f,k} = 1\right] + \frac{1}{2}\Pr\left[T_{a,b}^{f,k} = 1\right] > \Pr\left[D_{a,b}^{f_p,k} = 1\right] + \frac{1}{2}\Pr\left[T_{a,b}^{f_p,k} = 1\right]$ for all $a, b \in A$ such that $M[a, 1] > 0$ and $M[b, 1] > 0$;*

2. *If there exists $c \in A$ such that $M[c, 1] = 0$, then $\Pr\left[D_{a,c}^{f,k} = 1\right] + \frac{1}{2}\Pr\left[T_{a,c}^{f,k} = 1\right] \geq \Pr\left[D_{a,c}^{f_p,k} = 1\right] + \frac{1}{2}\Pr\left[T_{a,c}^{f_p,k} = 1\right]$ for all $a \in A$ and $k \in \mathbb{Z}^+$.*

Since $f_p(\boldsymbol{\sigma})$ contains no ties, there can be at most one $c \in A$ with $M[c, 1] = 0$. Since $m \geq 3$, the lemma implies that $\sum_{a,b \in A} \Pr\left[D_{a,b}^{f,k} = 1\right] + \frac{1}{2}\Pr\left[T_{a,b}^{f,k} = 1\right] > \sum_{a,b \in A} \Pr\left[D_{a,b}^{f_p,k} = 1\right] + \frac{1}{2}\Pr\left[T_{a,b}^{f_p,k} = 1\right]$ for all $k \geq k_f$. By definition of $AbC$, this implies that for all $k \geq k_f$ we have $f \notin AbC(F, \pi^k(\boldsymbol{\sigma}, [m-1]))$, since (4) is strictly greater for $f$ than for $f_p$. Since $|F|$ is finite, picking $k = \max_{f \in F \setminus \{f_p\}} k_f$ will ensure $AbC(F, \pi^{k'}(\boldsymbol{\sigma}, [m-1]) = \{f_p\}$ for all $k' \geq k$, as desired. We will now prove the lemma.

---

[4]If $m = 2$, there is a single (monotonic) positional scoring rule satisfying Definition 1: the one with score vector $(1, 0)$, which is also equivalent to $f_p$.

*Proof of Lemma 13.* For any $f \in F$, say $\{s_i^f\}_{i \in [m]}$ is the scoring vector associated with $f$. Fix any two candidates $a, b \in A$. Say $M[a, 1] = p_a$ and $M[b, 1] = p_b$, with $p_a > p_b$ WLOG.[5] By Definition 6, we have $M_k[a, 1] = kp_a m!$ and $M_k[b, 1] = kp_b m!$, whereas $M_k[a, i] = \frac{k(n-p_a)m!}{m-1}$ and $M_k[b, i] = \frac{k(n-p_b)m!}{m-1}$ for all $1 < i \le m$. Similarly, let $M_k[(a, i); (b, j)]$ be the number of voters in $\boldsymbol{\sigma}_k$ that rank $a$ in the $i$th position *and* $b$ in the $j$th position (with $i \ne j$). Then

$$M_k[(a, i); (b, j)] = \begin{cases} kp_a C_1 & \text{if } i = 1 \\ kp_b C_1 & \text{if } j = 1 \\ kC_2 & \text{otherwise} \end{cases},$$

where $C_1 \equiv \frac{m!}{m-1}$ and $C_2 \equiv \frac{(n-p_a-p_b)m!}{(m-1)(m-2)}$. Given a random split of $\boldsymbol{\sigma}_k$ into $\boldsymbol{\sigma}_k^{(1)}$ and $\boldsymbol{\sigma}_k^{(2)}$, say $X_i^{f,k}$ is the random variable indicating the total score of $a$ minus the total score of $b$ in $\boldsymbol{\sigma}_k^{(i)}$ according to $f$ for each $i \in \{1, 2\}$. Naturally, regardless of the split, $X_1^{f,k} + X_2^{f,k}$ must add up to the total score of $a$ minus the total score of $b$ in $\boldsymbol{\sigma}_k$ according to $f$, so

$$X_1^{f,k} + X_2^{f,k} = \sum_{i=1}^m s_i^f M_k[a, i] - \sum_{i=1}^m s_i^f M_k[b, i]$$

$$= kp_a m! - kp_b m! + \left( \frac{k(n-p_a)m!}{m-1} - \frac{k(n-p_b)m!}{m-1} \right) \sum_{i=2}^m s_i^f$$

$$= \frac{\lambda^f k(p_a - p_b)m!}{m-1}$$

where $\lambda^f = (m-1) - \sum_{i=2}^m s_i^f > 0$, as $s_m^f = 0$. Then, using symmetry of the two sides and the fact that $p_a > p_b$, we get

$$\Pr\left[ D_{a,b}^{f,k} = 1 \right] = \Pr\left[ X_1^{f,k} \cdot X_2^{f,k} < 0 \right] = \Pr\left[ X_1^{f,k} \cdot \left( \frac{\lambda^f k(p_a - p_b)m!}{m-1} - X_1^{f,k} \right) < 0 \right]$$

$$= \Pr\left[ X_1^{f,k} < 0 \right] + \Pr\left[ X_1^{f,k} > \frac{\lambda^f (p_a - p_b)m!}{m-1} \right]$$

$$= 2\Pr\left[ X_1^{f,k} < 0 \right].$$

A similar analysis gives $\Pr\left[ T_{a,b}^{f,k} = 1 \right] = 2\Pr\left[ X_1^{f,k} = 0 \right]$. Now, if $a$ and $b$ are ranked $i$th and $j$th in a ranking, respectively, the score assigned to $a$ minus that assigned to $b$ by $f$ for that ranking is $s_i^f - s_j^f$. Say $\text{Bin}(z)$ is the fair binomial distribution where $z$ is the number of experiments and the probability of success for each experiment is $1/2$. Then,

$$X_1^{f,k} \sim \sum_{i \ne j \in [m]} (s_i^f - s_j^f) \text{Bin}\left( M_k[(a, i); (b, j)] \right)$$

$$= \sum_{i < j \in [m]} (s_i^f - s_j^f) \left( \text{Bin}\left( M_k[(a, i); (b, j)] \right) - \text{Bin}\left( M_k[(a, j); (b, i)] \right) \right)$$

$$= \sum_{i < j \in [m]} (s_i^f - s_j^f) \left( \text{Bin}\left( M_k[(a, i); (b, j)] \right) + \text{Bin}\left( M_k[(a, j); (b, i)] \right) - M_k[(a, j); (b, i)] \right)$$

$$= \sum_{i < j \in [m]} (s_i^f - s_j^f) \left( \text{Bin}\left( M_k[(a, i); (b, j)] + M_k[(a, j); (b, i)] \right) - M_k[(a, j); (b, i)] \right)$$

$$= \sum_{i < j \in [2,m]} (s_i^f - s_j^f) \left( \text{Bin}\left( 2kC_2 \right) - kC_2 \right) + \sum_{i=2}^m (1 - s_i^f) \left( \text{Bin}\left( k(p_a + p_b)C_1 \right) - kp_b C_1 \right) \quad (5)$$

In particular, for plurality, equation 5 gives us:

$$X_1^{f_p,k} \sim \sum_{i=2}^m \left( \text{Bin}\left( k(p_a + p_b)C_1 \right) - kp_b C_1 \right) = \text{Bin}\left( k(p_a + p_b)m! \right) - kp_b m! \quad (6)$$

---

[5]We cannot have $p_a = p_b$ as by assumption $f_p(\boldsymbol{\sigma})$ contains no ties.

Now, fixing some $f \in F \setminus \{f_p\}$, using equation 5, we write $X_1^{f,k} = Y + \sum_{i=2}^{m}(1 - s_i^f)Z_i$, where

$$Y \sim \sum_{i<j \in [2,m]} (s_i^f - s_j^f)\left(\mathrm{Bin}\left(2kC_2\right) - kC_2\right), \text{ and}$$

$$Z_i \sim \left(\mathrm{Bin}\left(k(p_a + p_b)C_1\right) - kp_bC_1\right) \text{ for all } i \in [2,m],$$

each sampled independently. It is easy to check that $\mathbb{E}[Y] = 0$ and that $Y$ is a symmetric distribution. We will treat two cases separately.

**Case 1:** $p_b = 0$. In this case, we cannot have $X_1^{f_p,k} < 0$, as seen by (6). Therefore,

$$\Pr\left[D_{a,b}^{f_p,k} = 1\right] + \frac{1}{2}\Pr\left[T_{a,b}^{f_p,k} = 1\right] = 0 + \Pr\left[X_1^{f_p,k} = 0\right] = 2^{-kp_am!}. \tag{7}$$

For $f$, on the other hand, we have

$$\Pr\left[X_1^{f,k} = 0\right] \geq \Pr[Y = 0; Z_i = 0 \text{ for all } i \in [2,m]] = \frac{\Pr[Y = 0]}{(2^{kp_aC_1})^{m-1}} = \Pr[Y = 0] \cdot 2^{-kp_am!},$$

$$\text{and } \Pr\left[X_1^{f,k} < 0\right] \geq \Pr[Y < 0; Z_i = 0 \text{ for all } i \in [2,m]] = \frac{1 - \Pr[Y = 0]}{2} \cdot 2^{-kp_am!},$$

$$\Rightarrow \Pr\left[D_{a,b}^{f_p,k} = 1\right] + \frac{1}{2}\Pr\left[T_{a,b}^{f_p,k} = 1\right] = 2\Pr\left[X_1^{f,k} < 0\right] + \Pr\left[X_1^{f,k} = 0\right] \geq 2^{-kp_am!}. \tag{8}$$

Since $a \in A$ and $k \in \mathbb{Z}^+$ were arbitrarily chosen, combining (7) and (8) gives us statement 2 from the lemma.

**Case 2:** $p_b > 0$. Say $\phi(N, r)$ is $\Pr[A \leq rN]$ for $A \sim \mathrm{Bin}(N)$, given that $0 < r < \frac{1}{2}$ and $rN$ is an integer. By Ash (1990, Lemma 4.7.2), we know that

$$\frac{2^{-N(1-h(r))}}{\sqrt{8Nr(1-r)}} \leq \phi(N, r) \leq 2^{-N(1-h(r))}, \tag{9}$$

where $h(r) = -r \log r - (1-r)\log(1-r)$ is the binary entropy function. We will use equation 9 to upper bound the error probability for $f_p$ and lower bound the error probability for $f$. For convenience, define

$$r_p \stackrel{\text{def}}{=} \frac{p_b}{p_a + p_b}, \quad h_p \stackrel{\text{def}}{=} h(r_p), \quad N_p \stackrel{\text{def}}{=} (p_a + p_b)C_1.$$

First we start with $f_p$. By (6) and (9), we have

$$\Pr\left[D_{a,b}^{f_p,k} = 1\right] + \frac{1}{2}\Pr\left[T_{a,b}^{f_p,k} = 1\right] = 2\Pr\left[X_1^{f_p,k} < 0\right] + \Pr\left[X_1^{f_p,k} = 0\right]$$
$$\leq 2\Pr\left[X_1^{f_p,k} \leq 0\right] = 2\phi(k(p_a + p_b)m!, r_p)$$
$$\leq 2^{1-kN_p(m-1)(1-h_p)} \equiv U(k),$$

where we will use $U(k)$ as an upper bound that depends on $k$. Now consider $f$. Since $f \neq f_p$ and since $1 = s_1^f \geq s_2^f \geq \ldots s_m^f = 0$, we must have $s_2^f > 0$. Fix some $\varepsilon > 0$ such that $\varepsilon < \min\left(\frac{p_a - p_b}{2(p_a + p_b)}, \frac{p_b}{(p_a + p_b)(1 - s_2^f)}\right)$ (if $s_2^f = 1$, then ensuring $\varepsilon < \frac{p_a - p_b}{2(p_a + p_b)}$ is sufficient). Recalling that $X_1^{f,k} = Y + \sum_{i=2}^{m}(1 - s_i^f)Z_i$ and that $s_m^f = 0$, we have

$$\Pr\left[X_1^{f,k} \leq 0\right] \geq \Pr[Y \leq 0] \cdot \Pr[Z_2 \leq \varepsilon kN_p] \cdot \Pr\left[Z_m \leq -\varepsilon kN_p(1 - s_2^f)\right] \cdot \prod_{i=3}^{m-1}\Pr[Z_i \leq 0],$$

$$\Pr\left[X_1^{f,k} < 0\right] \geq \Pr[Y < 0] \cdot \Pr[Z_2 \leq \varepsilon kN_p] \cdot \Pr\left[Z_m \leq -\varepsilon kN_p(1 - s_2^f)\right] \cdot \prod_{i=3}^{m-1}\Pr[Z_i \leq 0]$$

$$= \Pr[Y > 0] \cdot \Pr[Z_2 \leq \varepsilon kN_p] \cdot \Pr\left[Z_m \leq -\varepsilon kN_p(1 - s_2^f)\right] \cdot \prod_{i=3}^{m-1}\Pr[Z_i \leq 0];$$

therefore,

$$
\begin{aligned}
\Pr\!\left[D_{a,b}^{f,k}=1\right]+\frac{1}{2}\Pr\!\left[T_{a,b}^{f,k}=1\right] &= 2\Pr\!\left[X_1^{f,k}<0\right]+\Pr\!\left[X_1^{f,k}=0\right]\\
&= \Pr\!\left[X_1^{f,k}<0\right]+\Pr\!\left[X_1^{f,k}\le 0\right]\\
&\ge \Pr[Z_2\le \varepsilon kN_p]\cdot \Pr\!\left[Z_m\le -\varepsilon kN_p(1-s_2^f)\right]\cdot \prod_{i=3}^{m-1}\Pr[Z_i\le 0]\\
&\ge \Pr[Z_2\le \lfloor \varepsilon kN_p\rfloor]\cdot \Pr\!\left[Z_m\le -\lceil \varepsilon kN_p(1-s_2^f)\rceil\right]\cdot \prod_{i=3}^{m-1}\Pr[Z_i\le 0]\\
&\ge \phi(kN_p,r_2(k))\cdot \phi(kN_p,r_m(k))\cdot (\phi(kN_p,r_p))^{m-3}
\end{aligned}
$$

where $r_2(k)=r_p+\frac{\lfloor \varepsilon kN_p\rfloor}{kN_p}$ and $r_m(k)=r_p-\frac{\lceil \varepsilon kN_p(1-s_2^f)\rceil}{kN_p}$. As $k\to\infty$, we have $r_2(k)\to r_p+\varepsilon\in\left(0,\frac{1}{2}\right)$ and $r_m(k)\to r_p-\varepsilon(1-s_2^f)\in\left(0,\frac{1}{2}\right)$; hence, we can choose $k$ large enough such that $0<r_i<1/2$ for both $i\in\{2,m\}$. Further, by construction, $kN_p\cdot r_i(k)$ is an integer for all $k\ge 0$ and $i\in\{2,m\}$. Define $h_i(k)\overset{\text{def}}{=}h_i(r_i(k))$ for $i\in\{2,m\}$. By equation 9, we then have

$$
\Pr\!\left[D_{a,b}^{f,k}=1\right]+\frac{1}{2}\Pr\!\left[T_{a,b}^{f,k}=1\right]\ge \phi(kN_p,r_2(k))\cdot \phi(kN_p,r_m(k))\cdot (\phi(kN_p,r_p))^{m-3}
$$

$$
\ge \frac{2^{-kN_p(1-h_2(k))}}{\sqrt{8kN_p r_2(k)(1-r_2(k))}}\cdot \frac{2^{-kN_p(1-h_m(k))}}{\sqrt{8kN_p r_m(k)(1-r_m(k))}}\cdot \left(\frac{2^{-kN_p(1-h_p)}}{\sqrt{8kN_p r_p(1-r_p)}}\right)^{m-3}
$$

$$
= \frac{2^{-kN_p(m-1-h_2(k)-h_m(k)-(m-3)h_p)}}{\sqrt{(8kN_p)^{m-1}r_2(k)(1-r_2(k))r_m(k)(1-r_m(k))(r_p(1-r_p))^{m-3}}}\equiv L(k),
$$

where we will use $L(k)$ as a lower bound that depends on $k$. Now, we would like to show that there exists $k_f^{a,b}$ such that $L(k)>U(k)$ for all $k\ge k_f^{a,b}$. We have

$$
\frac{L(k)}{U(k)}=\frac{2^{kN_p P(k)-1}}{\sqrt{(8kN_p)^{m-1}r_2(k)(1-r_2(k))r_m(k)(1-r_m(k))(r_p(1-r_p))^{m-3}}}, \tag{10}
$$

where $P(k)=(m-1)(1-h_p)-(m-1-h_2(k)-h_m(k)-(m-3)h_p)=h_2(k)+h_m(k)-2h_p$. We would like to show $\lim_{k\to\infty}\frac{L(k)}{U(k)}=\infty$, giving us our desired relationship. We recall that $r_2(k)\to r_p+\varepsilon$ and $r_2(k)\to r_p-\varepsilon(1-s_2^f)$ as $k\to\infty$, so the denominator of equation 10 scales as $\Theta(k^{\frac{m-1}{2}})$ for large $k$. As for the nominator, we have $\lim_{k\to\infty}P(k)=h(r_p+\varepsilon)+h(r_p-\varepsilon(1-s_2^f))-2h(r_p)\equiv H(\varepsilon)$. We have $H(0)=0$ and $H'(0)=h'(r_p)-(1-s_2^f)h'(r_p)=s_2^f h'(r_p)>0$ since $s_2^f>0$ and

$$
h'(r_p)=-\log r_p-1+\log(1-r_p)+1=\log\left(\frac{1}{r_p}-1\right)>\log(2-1)=0,
$$

as $r_p=\frac{p_b}{p_a+p_b}<\frac{1}{2}$. We can therefore set $\varepsilon>0$ small enough such that $H(\varepsilon)>0$. Then the nominator of equation 10 scales as $\Theta(2^{kN_p H(\varepsilon)})$ for large $k$, dominating the denominator. Hence, $\lim_{k\to\infty}\frac{L(k)}{U(k)}=\infty$, as desired. Therefore, we can choose a large enough $k_f^{a,b}$ such that for all $k\ge k_f^{a,b}$ we have

$$
\Pr\!\left[D_{a,b}^{f,k}=1\right]+\frac{1}{2}\Pr\!\left[T_{a,b}^{f,k}=1\right]\ge L(k)>U(k)\ge \Pr\!\left[D_{a,b}^{f_p,k}=1\right]+\frac{1}{2}\Pr\!\left[T_{a,b}^{f_p,k}=1\right].
$$

Since there are finitely many candidates $A$, picking $k_f=\max_{(a,b)\in A^2}k_f^{a,b}$ is sufficient for proving statement 1 from the lemma. $\square$

$\square$

**Proposition 14.** *No (anonymous) RPR can satisfy all three of reversal symmetry, plurality-shuffling consistency (PSC), and union consistency (UC).*

*Proof.* Assume an RPR $Z$ satisfies PSC and reversal symmetry. We will show that $Z$ necessarily fails UC. Fix $A = \{a, b, c\}$ and $F = \{f_p, f_v\}$. Say $\boldsymbol{\sigma}_a$ consists of two voters ranking $a \succ b \succ c$ and one voter ranking $b \succ a \succ c$. Similarly, say $\boldsymbol{\sigma}_b$ consists of two voters ranking $c \succ b \succ a$ and one voter ranking $b \succ c \succ a$. We have $f_p(\boldsymbol{\sigma}_a) = a \succ b \succ c$ and $f_p(\boldsymbol{\sigma}_b) = c \succ b \succ a$, so both profiles satisfy the preconditions of Definition 7. By PSC, there exist $k_a, k_b$ such that for all $k'_a \geq k_a$ and $k'_b \geq k_b$ we have $Z(F, \pi^{k'_a}(\boldsymbol{\sigma}_a, [2, m])) = Z(F, \pi^{k'_b}(\boldsymbol{\sigma}_b, [2, m])) = \{f_p\}$. Pick $k = \max\{k_a, k_b\}$ and let $\boldsymbol{\sigma} = \pi^k(\boldsymbol{\sigma}_a, [2, m]) + \pi^k(\boldsymbol{\sigma}_b, [2, m])$. It is straightforward to check that $\boldsymbol{\sigma} = \text{rev}(\boldsymbol{\sigma})$ up to a permutation of voters, with each $r \in \mathcal{L}(A)$ appearing $6k$ times. By anonymity and reversal symmetry, this implies $Z(F, \boldsymbol{\sigma}) = \text{rev}(Z(F, \boldsymbol{\sigma}))$, which is only possible if $Z(F, \boldsymbol{\sigma}) = \{f_v, f_p\}$, proving UC is violated. $\square$

### B.2 Proof of Theorem 2

**Theorem 2.** *(I) AbC preserves anonymity & neutrality. (II) Any RPR preserves the Smith criterion, Condorcet consistency, majority winner, pairwise majority consistency, and unanimity. (III) No (anonymous) RPR can satisfy reversal symmetry and preserve monotonicity.*

We prove each claim as a separate proposition.

**Proposition 15.** *AbC preserves anonymity & neutrality.*

*Proof.* The proof follows definitionally from the anonymity and neutrality of the Kendall-Tau distance (Definition 3), and therefore of the $AbC$ function (Box 1). $\square$

Next, we formally define the social choice axioms given in part (2) of Theorem 2. Given a profile $\boldsymbol{\sigma}$ and $a, b \in A$, we say $a$ pairwise defeats $b$ if $|\{i \in N : a \succ_{\sigma_i} b\}| > |\{i \in N : b \succ_{\sigma_i} a\}|$. Then, we say an SWF $f \in F$ satisfies. . .

- . . . the *Smith Criterion (SC)* if the alternative(s) ranked highest in $f(\boldsymbol{\sigma})$ belongs to the Smith set of $\boldsymbol{\sigma}$, *i.e.*, the smallest set $S \subseteq A$ such that every $a \in S$ pairwise defeats every $b \in A \backslash S$.

- . . . *Condorcet Consistency (CC)* if it satisfies the SC for all profiles $\boldsymbol{\sigma}$ that have a Smith set containing a single alternative (called the Condorcet winner).

- . . . *Majority Winner (MW)* if is satisfies CC whenever the Condorcet winner is the top-ranked candidate of a majority ($> m$) of voters.

- . . . *Pairwise Majority Consistency (PMC)* if whenever there exists $r \in \mathcal{R}(A)$ such that $a \succ_r b$ if and only if $a$ pairwise defeats $b$ in $\boldsymbol{\sigma}$, then $f(\boldsymbol{\sigma}) = r$. (Ge et al., 2024)

- . . . *unanimity* if whenever $\sigma_i = r$ for all $i \in N$, then $f(\boldsymbol{\sigma}) = r$.

**Proposition 16.** *Any RPR $Z$ preserves the Smith criterion, Condorcet consistency, majority winner, pairwise majority consistency, and unanimity.*

*Proof.* Like Proposition 16, the proof follows from the definitions of the axioms. Each of them are of the form "if $\boldsymbol{\sigma}$ satisfies conditions $X$, then $f(\boldsymbol{\sigma})$ must satisfy conditions $Y$." Since for each $\boldsymbol{\sigma}$ we have $f_Z^F(\boldsymbol{\sigma}) = f(\boldsymbol{\sigma})$ for some $f \in F$, restricting $F$ to rules that satisfy this axiom will ensure $f_Z^F(\boldsymbol{\sigma})$ will satisfy $Y$ whenever $\boldsymbol{\sigma}$ satisfies $X$. $\square$

Next, we formally define monotonicity.

**Definition 17.** Given a weak ranking $r \in \mathcal{R}(A)$ and an alternative $a \in A$, say $\text{rank}_r(a) = 1 + |\{b \in A : b \succ_r a\}|$ is the rank of $a$ in $r$. We say an SWF $f$ satisfies *monotonicity* if for any profile $\boldsymbol{\sigma}$ and alternative $a \in A$, if $\boldsymbol{\sigma}'$ is the same as $\boldsymbol{\sigma}$ except some voters now rank $a$ higher, then $\text{rank}_{f(\boldsymbol{\sigma})}(a) \geq \text{rank}_{f(\boldsymbol{\sigma}')}(a)$.

Before proving the general impossibility result in part (3) of Theorem 2, we will show that $AbC$ does not preserve monotonicity.

**Example 18.** *Say $F = \{f_p, f_v\}$ (plurality and veto). Both of these rules are monotonic, as all positional scoring rules are. Fix $k \in \mathbb{Z}^+$ and consider the following profile $\boldsymbol{\sigma}$:*

- *Group 1: $k$ voters rank $a \succ b \succ c \succ d$.*

- *Group 2: $k$ people voted $d \succ c \succ a \succ b$.*

- *Group 3: $k$ people voted $d \succ c \succ a \succ b$.*

- *Group 4: $k$ voters rank $b \succ a \succ c \succ d$.*

- *Group 5: $3k$ voters rank $c \succ a \succ b \succ d$.*

- *Group 6: $3k$ voters rank $d \succ b \succ a \succ c$.*

*The veto scores (the number of voters that rank them bottom) of $a, b, c, d$ are $0, 2k, 3k, 5k$, respectively. By an analogous argument to that in Example 4, $\Pr\big[f_v(\boldsymbol{\sigma}^{(1)}) = f_v(\boldsymbol{\sigma}^{(2)}) = (a \succ b \succ c \succ d)\big] \rightarrow 1$ as $k \rightarrow \infty$, where the probabilities are taken over the random process in Box 1. Hence, the expectation in Equation (1) of Box 1 approaches 0 for $f_v$ as $k$ grows. However, the plurality scores (the number of voters that rank them top) of $a$ and $b$ are tied, implying the the probability that $f_p(\boldsymbol{\sigma}^{(1)})$ and $f_p(\boldsymbol{\sigma}^{(2)})$ will disagree about $a$ and $b$ converges to 1 as $k \rightarrow \infty$. This implies the expectation in (1) approaches at least 1 for $f_p$ as $k$ grows. Therefore, there exists a $k_1$ such that $AbC(F, \boldsymbol{\sigma}) = \{f_v\}$ for all $k \geq k_1$, and therefore $f_{AbC}^F(\boldsymbol{\sigma}) = (a \succ b \succ c \succ d)$. Now say the voters in group 1 and 2 promote $b$ by a single spot, to get the profile $\boldsymbol{\sigma}'$:*

- *Group 1: $k$ voters rank $b \succ a \succ c \succ d$.*

- *Group 2: $k$ people voted $d \succ c \succ b \succ a$.*

- *Group 3: $k$ people voted $d \succ c \succ a \succ b$.*

- *Group 4: $k$ voters rank $b \succ a \succ c \succ d$.*

- *Group 5: $3k$ voters rank $c \succ a \succ b \succ d$.*

- *Group 6: $3k$ voters rank $d \succ b \succ a \succ c$.*

*Importantly, we have $\boldsymbol{\sigma}' = \text{rev}(\boldsymbol{\sigma})$ up to permuting Groups 1 with 2, 3 with 4, and 5 with 6. By anonymity and reversal symmetry, this implies $AbC(F, \boldsymbol{\sigma}') = \{f_p\}$, so $f_{AbC}^F(\boldsymbol{\sigma}') = (d \succ c \succ b \succ a)$. Thus, by promoting $b$, we have gotten $\text{rank}_{f_{AbC}^F(\boldsymbol{\sigma})}(b) = 2 < 3 = \text{rank}_{f_{AbC}^F(\boldsymbol{\sigma}')}(b)$, which violates monotonicity.*

We now show how the same example can be used for other (anonymous) RPRs satisfying reversal symmetry, in order to prove part (3) of Theorem 2.

**Proposition 19.** *No (anonymous) RPR can satisfy reversal symmetry and preserve monotonicity.*

*Proof.* Say RPR $Z$ satisfies reversal symmetry and is anonymous. Take $\boldsymbol{\sigma}, \boldsymbol{\sigma}'$ and $F$ from Example 18. We consider two cases.
**Case 1:** $Z(F, \boldsymbol{\sigma}) = \{f_v\}$. Then, by the same reasoning as in the proof of Example 18, we must have $Z(F, \boldsymbol{\sigma}') = \{f_p\}$ (as that example only uses anonymity and reversal symmetry to show this), and monotonicity is violated by $f_Z^F$.
**Case 2:** $Z(F, \boldsymbol{\sigma}) = \{f_p\}$. In that case, $f_Z^F(\boldsymbol{\sigma}) = (d \succ c \succ a = b)$, *i.e.*, $a$ and $b$ are tied in the bottom. By reversal symmetry, we have $Z(F, \boldsymbol{\sigma}') = \{f_v\}$ and therefore $f_Z^F(\boldsymbol{\sigma}') = (a = b \succ c \succ d)$. However, one can go from $\boldsymbol{\sigma}'$ to $\boldsymbol{\sigma}$ by simply promoting $a$ by a spot in the rankings of groups 1 and 2. This implies that $\text{rank}_{f_Z^F(\boldsymbol{\sigma}')}(a) = 1 < 3 = \text{rank}_{f_Z^F(\boldsymbol{\sigma})}(a)$, so promoting $a$ results in increasing its rank, therefore $f_Z^F$ is not monotonic.

In either case, we see that $f_Z^F$ violates monotonicity. Since $f_p$ and $f_v$ are both monotonic, this proves that $Z$ does not preserve monotonicity. $\qquad\square$

## B.3 PROOF OF THEOREM 3

We recall the complexity result from Section 6.

**Theorem 3.** PERFPOS *is* NP-*complete.*

We first introduce a generalization of the computational problem PERFPOS, which we will call $k$PERFPOS: For each voter $i \in N$, we are given a strict ranking $\sigma_i \in \mathcal{L}(A_i)$ over a *subset* of alternatives $A_i \subseteq A$, with $|A_i| = k$. We are also given a split of voters $N = N_1 \sqcup N_2$ with $|N_1| = |N_2|$. For each $j \in \{1, 2\}$, $a \in A$, and $i \in [k]$, $M_j[a, i]$ indicates the number of voters in $N_j$ that rank $a$ in their $i^{\text{th}}$ position. Then, PERFPOS asks: is there a positional scoring rule $f_s \in F_S$ that achieves zero disagreement over this split, *i.e.*, is there a vector $s = (s_i)_{i \in [k]}$ with $1 = s_1 \geq s_2 \geq \ldots \geq s_k = 0$ such that for all $a, b \in A$

$$(T_1[a] - T_1[b])(T_2[a] - T_2[b]) > 0,$$

where $T_j[a] = \sum_{i=1}^{k} M_j[a, i]s_i$ for any $a \in A$ and $j \in \{1, 2\}$.

Clearly $k$PERFPOS contains PERFPOS (for $k = m$). However, as we show next, it is not harder: Given an instance of $k$PERFPOS (with input profile $\boldsymbol{\sigma}$), complete the ranking of each voter $i \in N$ to a strict ranking over all alternatives by placing the remaining $m - k$ alternatives in $A \setminus A_i$ at the bottom of $\sigma_i$, giving rise to a new (complete) profile $\boldsymbol{\sigma}'$. Then define $\boldsymbol{\sigma}'' = \pi^1(\boldsymbol{\sigma}', [k, m])$, *i.e.*, the $1-$shuffling of $\boldsymbol{\sigma}'$ with respect to positions $[k, m] = \{k, k+1, \ldots, m\}$ (Definition 6). Then, we claim the answer to the original $k$PERFPOS instance with profile $\boldsymbol{\sigma}$ is a yes if and only if the answer to the PERFPOS instance using $\boldsymbol{\sigma}''$ (and the same split as $\boldsymbol{\sigma}$) is a yes. Indeed, if there exists a positional scoring rule $f_s$ with $1 = s_1 \geq s_2 \geq \ldots \geq s_k = 0$ that achieves zero disagreement with $\boldsymbol{\sigma}$, then $f_{s''}$ with $s'' = (s_i)_{i \in [m]}$ defined as $s_i'' = \begin{cases} s_i & \text{if } i \leq k \\ 0 & \text{otherwise} \end{cases}$ achieves zero disagreement with $\boldsymbol{\sigma}''$. On the other hand, given a positional scoring rule $f_{s''}$ with $1 = s_1'' \geq s_2'' \geq \ldots \geq s_m'' = 0$ that achieves zero disagreement with $\boldsymbol{\sigma}''$, define $x = \frac{s_k'' + s_{k+1}'' + \ldots + s_m''}{m - k + 1}$ and define vector $s = (s_i)_{i \in [k]}$ as $s_i = (s_i'' - x)/(1 - x)$ for all $i \in [k-1]$ and $s_k = 0$ (We have $x < 1$ since $s_m'' = 0$). We will show $f_s$ achieves zero disagreement with $\boldsymbol{\sigma}$. Given any $a \in A$ and $j \in \{1, 2\}$, the total score assigned by $f_{s''}$ to $a$ on input $\boldsymbol{\sigma}''^{(j)}$ (restriction of $\boldsymbol{\sigma}''$ to voters in $N_j$) is

$$T_j''[a] = \sum_{i=1}^{k-1} k! M_j[a, i]s_i'' + \sum_{i=k}^{m} k! \frac{|N_j| - \sum_{i'=1}^{k-1} M_j[a, i']}{m - k + 1} s_i''$$

$$= k! \left( x|N_j| + \sum_{i=1}^{k-1} M_j[a, i](s_i'' - x) \right)$$

by Definition 6. The score assigned by $f_s$ to $a$ on input $\boldsymbol{\sigma}^{(j)}$ (restriction of $\boldsymbol{\sigma}$ to voters in $N_j$), on the other hand, is

$$T_j[a] = \sum_{i=1}^{k} M_j[a, i]s_i = \sum_{i=1}^{k-1} M_j[a, i] \frac{s_i'' - x}{1 - x} = \frac{T_j''[a] - k!x|N_j|}{k!(1 - x)}.$$

Since the total score of every alternative is just shifted by a constant and then rescaled by another constant, for any $a, b \in A$ we have $T_j''[a] > T_j''[b]$ if and only if $T_j[a] > T_j[b]$. As $f_{s''}$ achieves zero disagreement with $\boldsymbol{\sigma}''$, this proves $f_s$ achieves zero disagreement with $\boldsymbol{\sigma}$. Thus, in the proof of Proposition 20 below, we reduce 3SAT to $k$PERFPOS, which proves the NP-hardness for PERFPOS too (membership follows from the fact that positional scoring rules are easy to compute).

**Proposition 20.** $k$PERFPOS *is* NP-*hard.*

*Proof.* We will be reducing from 3SAT. Say $\phi$ is a 3CNF formula with clauses $C_1, C_2, \ldots, C_\ell$ and binary variables $x_1, x_2, \ldots, x_t$. For each $\beta \in \{1, 2\}$, we will construct and instance of PERFPOS by first specifying $M_\beta[a, i]$ for alternative $a \in A$ and position $i \in [k]$, and then explicitly designing rankings that is consistent with that $M_\beta$. We start with setting $k = t + 2$ and $\varepsilon = \frac{1}{7(k+2)}$.

For each $i \in [t]$, add two candidates $a_i$ and $b_i$, with

$$M_1[a_i, j] = \begin{cases} 1 + (k+3)(i-1) & \text{if } j = 1 \\ k+2 & \text{if } j = i+1 \\ 0 & \text{otherwise} \end{cases}$$

$$M_1[b_i, j] = \begin{cases} (k+3)(i-1) & \text{if } j = 1 \text{ and } i \neq 1 \\ k+2 & \text{if } j = i \\ 1 & \text{if } j = k \\ 0 & \text{otherwise} \end{cases}$$

and

$$M_2[a_i, j] = \begin{cases} 1 + (1/\varepsilon + 1)(i-1) & \text{if } j = 1 \\ 1/\varepsilon & \text{if } j = i+1 \\ 0 & \text{otherwise} \end{cases}$$

$$M_2[b_i, j] = \begin{cases} (1/\varepsilon + 1)(i-1) & \text{if } j = 1 \text{ and } i \neq 1 \\ 1/\varepsilon & \text{if } j = i \\ 1 & \text{if } j = k \\ 0 & \text{otherwise} \end{cases}.$$

For each $i \in [\ell]$, say the clause $C_i$ in $\phi$ consist of variables $\{x_j\}_{j \in V_i}$ for $V_i \subseteq [t]$, with $1 \leq |V_i| \leq 3$. For this clause $C_i$, add two candidates $c_j, d_j$, with

$$M_1[c_i, j] = \begin{cases} t(k+3) + 2i & \text{if } j = 1 \\ 0 & \text{otherwise} \end{cases} \qquad M_1[d_i, j] = \begin{cases} t(k+3) + 2i - 1 & \text{if } j = 1 \\ 1 & \text{if } j = k \\ 0 & \text{otherwise} \end{cases}.$$

For constructing $M_2[c_i, j]$ and $M_2[d_i, j]$, say $z \in \{0, 1, 2, 3\}$ is the number of negated literals in $C_i$. To build $M_2[c_i, j]$ and $M_2[d_i, j]$ for each $j \in [k]$, start from the following:

$$M_2'[c_i, j] = \begin{cases} 2z + t(1/\varepsilon + 1) + 6(k+3)(i-1) & \text{if } j = 1 \\ 1 & \text{if } j = k \\ 0 & \text{otherwise} \end{cases}$$

$$M_2'[d_i, j] = \begin{cases} 1 + t(1/\varepsilon + 1) + 6(k+3)(i-1) & \text{if } j = 1 \\ 2z & \text{if } j = k \\ 0 & \text{otherwise} \end{cases}.$$

Now for each $j \in V_i$, do the following:

- If $x_j$ appears non-negated in $C_i$, then add $2(k+2)$ to $M_2'[c_i, j]$ and $2(k+2)$ to $M_2'[d_i, j+1]$.

- If $x_j$ appears negated in $C_i$, then add $2(k+2)$ to $M_2'[c_i, j+1]$ and $2(k+2)$ to $M_2'[d_i, j]$.

Finally, set $M_2[c_i, j]$ and $M_2[d_i, j]$ to the resulting $M_2'[c_i, j]$ and $M_2'[d_i, j]$ for each $j \in [k]$.

Say $A = \{a_i\}_{i \in [t]}, B = \{b_i\}_{i \in [t]}, C = \{c_i\}_{i \in [\ell]}$, and $D = \{d_i\}_{i \in [\ell]}$. We now construct a profile that corresponds to the above $M_\beta$. First, add $k$ more candidates $E = \{e_i\}_{i \in [k]}$. Now for each $\beta \in \{1, 2\}$, each $f \in A \sqcup B \sqcup C \sqcup D$ and each $i \in [k]$, add $M_\beta[f, i]$ voters to the set $N_\beta$ that rank $f$ in the $i^{\text{th}}$ position, and ranks $e_j$ in the $j^{\text{th}}$ position for all $j \in [k] \setminus \{i\}$. Say we have added $n_1$ and $n_2$ voters to $N_1$ and $N_2$ so far respectively and that $\beta' = \arg\max_{\beta \in \{1,2\}} n_\beta$ (If it's a tie, pick $\beta'$ it arbitrarily). Add $(14k + 28)n_{\beta'}$ and $(14k + 29)n_{\beta'} - n_{3-\beta'}$ voters to $N_{\beta'}$ and $N_{3-\beta'}$, respectively, all of whom rank $e_i$ in the $i^{\text{th}}$ position for all $i \in [k]$. This also ensures that $|N_1| = |N_2|$. For each $\beta \in \{1, 2\}$ say $\boldsymbol{\sigma}^{(\beta)}$ is the final vector of rankings of $N_\beta$, as specified.

By construction, for each $\beta \in \{1, 2\}$, $f \in A \sqcup B \sqcup C \sqcup D$, and $i \in [k]$, $\boldsymbol{\sigma}^{(\beta)}$ indeed has $M_\beta[f, i]$ voters that rank $f$ in their $i^{\text{th}}$ position. Further, we have a total of $|A \sqcup B \sqcup$

$C \sqcup D \sqcup E| = 2(t + \ell) + k = 3t + 2\ell + 2$ candidates (which polynomial in $t, \ell$), and $|N_1| = |N_2| = (14k + 29) \max_{\beta \in \{1,2\}} \sum_{f \in A \cup B \cup C \cup D} \sum_{i=1}^{k} M_\beta[f, i] \le (14k + 29)2(t + \ell)k \cdot \max_{f \in A \cup B \cup C \cup D, i \in [k]} M_\beta[f, i]$ (which is also polynomial in $t, \ell$ since all entries of $M_\beta[f, i]$ are).

We now claim that $\phi$ is satisfiable if and only if there exists a positional scoring rule that gives full agreement between $\sigma^{(1)}$ and $\sigma^{(2)}$.

$(\Leftarrow)$ : Assume there is a positional scoring rule $f_s$ with $s = (s_i)_{i \in k}$ that gives full agreement for $M_1$ and $M_2$. Say $T_\beta[a] = \sum_{i=1}^{k} M_\beta[a, i]s_i$ for all $a \in A$ and $\beta \in \{1, 2\}$. In particular, we must have agreement between $a_i$ and $b_i$ for each $i \in [t]$. We have

$$T_1[a_i] = (1 + (k + 3)(i - 1))s_1 + (k + 2)s_{i+1}, \quad T_1[b_i] = ((k + 3)(i - 1))s_1 + (k + 2)s_i + s_k,$$
$$T_2[a_i] = (1 + (1/\varepsilon + 1)(i - 1))s_1 + s_{i+1}/\varepsilon, \quad T_2[b_i] = ((1/\varepsilon + 1)(i - 1))s_1 + s_i/\varepsilon + s_k.$$

Since $s_1 = 1$ and $s_m = 0$, perfect agreement implies that we must have

$$(T_1[a_i] - T_1[b_i])(T_2[a_i] - T_2[b_i]) = (1 - (k + 2)(s_i - s_{i+1}))(1 - (s_i - s_{i+1})/\varepsilon) > 0.$$

This implies we must either have $s_i - s_{i+1} < \varepsilon$ or $s_i - s_{i+1} > \frac{1}{k+2}$. Set the binary variable $x_i$ to False if $s_i - s_{i+1} < \varepsilon$ and to True if $s_i - s_{i+1} > \frac{1}{k+2}$. We now argue that the resulting $\{x_i\}_{i \in [t]}$ satisfies $\phi$, $i.e.$, satisfies all of its clauses. Fix any $i \in [\ell]$. We will show that $C_i$ is satisfied. By assumption of full agreement,

$$(T_1[c_i] - T_1[d_i])(T_2[c_i] - T_2[d_i]) > 0.$$

Since $T_1[c_i] - T_1[d_i] = (t(k + 3) + 2i)s_1 - (t(k + 3) + 2i - 1)s_1 - s_k = 1 > 0$, this implies that $(T_2[c_i] - T_2[d_i]) > 0$. Say $\{x_j\}_{j \in V_i}$ are the variables that appear in $C_i$ (for $V_i \subseteq [t]$), and that $Z = \{j \in V_i : x_j$ is negated in $C_i\}$ with $|Z| = z$. This implies we have

$$T_2[c_i] = (2z + t(1/\varepsilon + 1) + 6(k + 3)(i - 1))s_1 + s_k + 2(k + 2)\left(\sum_{g \in V_i \setminus Z} s_g + \sum_{g \in Z} s_{g+1}\right),$$

$$T_2[d_i] = (1 + t(1/\varepsilon + 1) + 6(k + 3)(i - 1))s_1 + 2zs_k + 2(k + 2)\left(\sum_{g \in V_i \setminus Z} s_{g+1} + \sum_{g \in Z} s_g\right),$$

$$T_2[c_i] - T_2[d_i] = 2z - 1 + 2(k + 2)\left(\sum_{g \in V_i \setminus Z}(s_g - s_{g+1}) - \sum_{g \in Z}(s_g - s_{g+1})\right).$$

The only way for $C_i$ to be not satisfied is if $x_g$ is assigned to True ($i.e.$, $s_g - s_{g+1} > \frac{1}{k+2}$) for all $g \in Z$ and $x_h$ is assigned to False ($i.e.$, $s_h - s_{h+1} < \varepsilon$) for all $h \in V_i \setminus Z$. This would imply, however $T_2[c_i] - T_2[d_i] < 2z - 1 + 2(k + 2)\left(\sum_{g \in V_i \setminus Z} \varepsilon - \sum_{g \in Z} \frac{1}{k+2}\right) = 2z - 1 + \frac{2(|V_i| - z)}{7} - 2z < -1 + \frac{6}{7} < 0$, which gives a contradiction.

As assuming $C_i$ is not satisfied gives a contradiction to the assumption that $s$ gives agreement between $\sigma^{(1)}$ and $\sigma^{(2)}$ for $c_i$ and $d_i$, $C_i$ must be satisfied. Since this is true for all $i \in [\ell]$, this implies that $\phi$ is satisfiable.

$(\Rightarrow)$ : Assume $\phi$ is satisfiable for truth assignments $\{x_i^*\}_{i \in [t]}$. For any $i \in [t]$, define $\delta_i = \begin{cases} \frac{1}{k+1} & \text{if } x_i^* \text{ is True} \\ \frac{\varepsilon}{2} & \text{otherwise.} \end{cases}$. Define the positional scoring rule $s = (s_i)_{i \in [k]}$ as $s_i = \begin{cases} 1 & \text{if } i = 1 \\ 1 - \left(\sum_{j=1}^{i-1} \delta_j\right) & \text{if } k > i > 1 \\ 0 & \text{if } i = k \end{cases}$. Since $s_{k-1} = 1 - \left(\sum_{j=1}^{k-2} \delta_j\right) = \ge 1 - \frac{k-2}{k+1} > 0 = s_k$, mono-

tonicity is satisfied, and $s$ is a valid scoring rule. We will show that $s$ gives perfect agreement between $\sigma^{(1)}$ and $\sigma^{(2)}$. As always, say $T_\beta[f] = \sum_{i=1}^{k} M_\beta[f, j]s_i$ for all $\beta \in \{1, 2\}$ and $f \in A$. Since $1 = s_1 \ge s_2 \ge \ldots \ge s_k = 0$ and since $s_i > 0$ for each $i \in [k - 1]$, the total scores for each

$i \in [t]$ and $j \in [\ell]$ are

$$T_1[a_i] = (1 + (k+3)(i-1))s_1 + (k+2)s_{i+1} \Rightarrow (k+3)i \geq T_1[a_i] > (k+3)(i-1) \quad \text{(A1)}$$

$$T_2[a_i] = (1 + (1/\varepsilon + 1)(i-1))s_1 + (1/\varepsilon)s_{i+1} \Rightarrow (1/\varepsilon + 1)i \geq T_2[a_i] > (1/\varepsilon + 1)(i-1) \quad \text{(A2)}$$

$$T_1[b_i] = (k+3)(i-1)s_1 + (k+2)s_i + s_k \Rightarrow (k+3)i > T_1[b_i] > (k+3)(i-1) \quad \text{(B1)}$$

$$T_2[b_i] = (1/\varepsilon + 1)(i-1)s_1 + (1/\varepsilon)s_i + s_k \Rightarrow (1/\varepsilon + 1)i > T_2[b_i] > (1/\varepsilon + 1)(i-1) \quad \text{(B2)}$$

$$T_1[c_j] = (t(k+3) + 2j)s_1 \Rightarrow t(k+3) + 2j = T_1[c_j] > t(k+3) + 2(j-1) \quad \text{(C1)}$$

$$T_2[c_j] = (2|Z_j| + t(1/\varepsilon + 1) + 6(k+3)(j-1))s_1 + s_k + 2(k+2)\left( \sum_{g \in V_j \setminus Z_j} s_g + \sum_{g \in Z_j} s_{g+1} \right)$$

$$\Rightarrow t(1/\varepsilon + 1) + 6(k+3)j \geq T_2[c_j] > t(1/\varepsilon + 1) + 6(k+3)(j-1) \quad \text{(C2)}$$

$$T_1[d_j] = (t(k+3) + 2j - 1)s_1 + s_k \Rightarrow t(k+3) + 2j > T_1[d_j] > t(k+3) + 2(j-1) \quad \text{(D1)}$$

$$T_2[d_j] = (1 + t(1/\varepsilon + 1) + 6(k+3)(j-1))s_1 + 2|Z_j|s_k + 2(k+2)\left( \sum_{g \in V_j \setminus Z_j} s_{g+1} + \sum_{g \in Z_j} s_g \right)$$

$$\Rightarrow t(1/\varepsilon + 1) + 6(k+3)j > T_2[d_j] > t(1/\varepsilon + 1) + 6(k+3)(j-1) \quad \text{(D2)}$$

where $V_j \subseteq [t]$ and $Z_j \subseteq V_j$ indicate the indices of the variables that appear in and that appear negated in the clause $C_j$, respectively. We now show that $s$ gives perfect agreement between $\boldsymbol{\sigma}^{(1)}$ and $\boldsymbol{\sigma}^{(2)}$, i.e., $(T_1[g] - T_1[h])(T_2[g] - T_2[h]) > 0$ for all distinct pairs of $g, h \in A \sqcup B \sqcup C \sqcup D \sqcup E$. We will proceed by a case by case analysis:

- **Case 1:** $g \in A \sqcup B$, $h \in C \sqcup D$. By eqs. (A1) and (B1), we have $T_1[g] \leq t(k+3)$, since $i \leq t$. By eqs. (C1) and (D1), we have $T_1[h] > t(k+3)$, as $j \geq 1$. This implies $T_1[g] < T_1[h]$. Similarly, by eqs. (A2) and (B2) we have $T_2[g] \leq t(1/\varepsilon + 1)$, and by eqs. (C2) and (D2) we have $T_2[h] > t(1/\varepsilon + 1)$, as $j \geq 1$. This implies $T_2[g] < T_2[h]$. Hence, $(T_1[g] - T_1[h])(T_2[g] - T_2[h]) > 0$, as desired.

- **Case 2:** $g \in \{a_i, b_i\}$, $h \in \{a_j, b_j\}$ for some $t \geq i > j \geq 1$. By eqs. (A1) and (B1), we have $T_1[g] > (k+3)(i-1) \geq (k+3)j$ and $T_1[h] \leq (k+3)j$. This implies $T_1[g] > T_1[h]$. Similarly, by eqs. (A2) and (B2) we have $T_2[g] > (1/\varepsilon + 1)(i-1) \geq (1/\varepsilon + 1)j$ and $T_2[h] \leq (1/\varepsilon + 1)j$. This implies $T_2[g] > T_2[h]$. Hence, $(T_1[g] - T_1[h])(T_2[g] - T_2[h]) > 0$, as desired.

- **Case 3:** $g = a_i$, $h = b_i$ for some $i \in [t]$. In this case $T_1[g] - T_1[h] = 1 - (k+2)(s_i - s_{i+1}) = 1 - (k+2)\delta_i$ and $T_2[g] - T_2[h] = 1 - (1/\varepsilon)(s_i - s_{i+1}) = 1 - \delta_i/\varepsilon$. If $\delta_i = \frac{\varepsilon}{2}$ (i.e. $x_i^*$ is False), $(T_1[g] - T_1[h])(T_2[g] - T_2[h]) = (1 - \frac{1}{14})(1 - \frac{1}{2}) > 0$. If $\delta_i = \frac{1}{k+1}$ (i.e. $x_i^*$ is True), $(T_1[g] - T_1[h])(T_2[g] - T_2[h]) = (1 - \frac{k+2}{k+1})(1 - \frac{7(k+2)}{k+1}) = (\frac{-1}{k+1})(\frac{-6k-13}{k+1}) > 0$.

- **Case 4:** $g \in \{c_i, d_i\}$, $h \in \{c_j, d_j\}$ for some $t \geq i > j \geq 1$. By eqs. (C1) and (D1), we have $T_1[g] > t(k+3) + 2(i-1) \geq t(k+3) + 2j$ and $T_1[h] \leq t(k+3) + 2j$. This implies $T_1[g] > T_1[h]$. Similarly, by eqs. (C2) and (D2) we have $T_2[g] > t(1/\varepsilon + 1) + 6(k+3)(i-1) \geq t(1/\varepsilon + 1) + 6(k+3)j$ and $T_2[h] \leq t(1/\varepsilon + 1) + 6(k+3)j$. This implies $T_2[g] > T_2[h]$. Hence, $(T_1[g] - T_1[h])(T_2[g] - T_2[h]) > 0$, as desired.

- **Case 5:** $g = c_i$, $h = d_i$ for some $i \in [\ell]$. In this case $T_1[g] - T_1[h] = 1$ and

$$T_2[g] - T_2[h] = 2|Z_i| - 1 + 2(k+2)\left( \sum_{g \in V_i \setminus Z_i} (s_g - s_{g+1}) - \sum_{h \in Z_i} (s_h - s_{h+1}) \right)$$

$$= 2|Z_i| - 1 + 2(k+2)\left( \sum_{g \in V_i \setminus Z_i} \delta_g - \sum_{h \in Z_i} \delta_h \right),$$

where $V_i \subseteq [t]$ and $Z_i \subseteq V_i$ indicate the indices of the variables that appear in and that appear negated in the clause $C_i$, respectively. Since $\{x_i^*\}_{i \in t}$ is a satisfying assignment by assumption, we must either have $\delta_{g'} = \frac{1}{k+1}$ (i.e. $x_{g'}^*$ is True) for some $g' \in V_i \setminus Z_i$ or $\delta_{h'} = \frac{\varepsilon}{2}$ (i.e. $x_{h'}^*$ is False) for some $h' \in Z_i$. If the former is true ($\delta_{g'} = \frac{1}{k+1}$ for some $g' \in V_i \setminus Z_i$):

$$
\begin{aligned}
T_2[g] - T_2[h] &= 2|Z_i| - 1 + 2(k+2)\left(\frac{1}{k+1} + \sum_{g \in V_i \setminus (Z_i \cup \{g'\})} \delta_g - \sum_{h \in Z_i} \delta_h\right) \\
&\geq 2|Z_i| - 1 + 2(k+2)\left(\frac{1}{k+1} + \frac{(|V_i| - |Z_i| - 1)\varepsilon}{2} - \frac{|Z_i|}{k+1}\right) \\
&\geq 2|Z_i| - 1 + 2(k+2)\left(\frac{1 - |Z_i|}{k+1} - \frac{\varepsilon}{2}\right) \\
&= -\frac{2|Z_i|}{k+1} + \frac{k+3}{k+1} - \frac{1}{7} = \frac{-2|Z_i| + \frac{6}{7}k + \frac{20}{7}}{k+1}.
\end{aligned}
$$

Similarly, if the latter is true ($\delta_{h'} = \frac{\varepsilon}{2}$ for some $h' \in Z_i$):

$$
\begin{aligned}
T_2[g] - T_2[h] &= 2|Z_i| - 1 + 2(k+2)\left(\sum_{g \in V_i \setminus Z_i} \delta_g - \frac{\varepsilon}{2} - \sum_{h \in Z_i \setminus \{h'\}} \delta_h\right) \\
&\geq 2|Z_i| - 1 + 2(k+2)\left(-\frac{\varepsilon}{2} - \frac{|Z_i| - 1}{k+1}\right) = -\frac{2|Z_i|}{k+1} + \frac{k+3}{k+1} - \frac{1}{7} \\
&= \frac{-2|Z_i| + \frac{6}{7}k + \frac{20}{7}}{k+1},
\end{aligned}
$$

which gives the same inequality. Consider two cases: if $t \geq 2$ (i.e., $k = t + 2 \geq 4$), we get

$$
T_2[g] - T_2[h] \geq \frac{-2|Z_i| + \frac{6}{7}k + \frac{20}{7}}{k+1} \geq \frac{-6 + \frac{24}{7} + \frac{20}{7}}{k+1} = \frac{\frac{2}{7}}{k+1} > 0,
$$

since $|Z_i| \leq 3$. If $t = 1$ (i.e., $k = 3$), then $|Z_i| \leq 1$, since $t$ is the number of variables in $\phi$. Then,

$$
T_2[g] - T_2[h] \geq \frac{-2|Z_i| + \frac{6}{7}k + \frac{20}{7}}{k+1} \geq \frac{-2 + \frac{18}{7} + \frac{20}{7}}{k+1} = \frac{\frac{24}{7}}{k+1} > 0.
$$

In both cases, we have $T_2[g] - T_2[h] > 0$ and hence $(T_1[g] - T_1[h])(T_2[g] - T_2[h]) > 0$, as desired.

- **Case 6:** $g \in E \setminus \{e_k\}, h \notin E$. Say $g = e_i$ for some $1 \leq i < k$ and $n_\beta = \sum_{f \in A \sqcup B \sqcup C \sqcup D} \sum_{j=1}^{k} M_\beta[f, j]$ for $\beta \in \{1, 2\}$. Due to the last set of voters that we added while constructing $\boldsymbol{\sigma}^{(1)}$ and $\boldsymbol{\sigma}^{(2)}$ (those that rank only elements of $E$), we have

$$
\begin{aligned}
M_\beta[e_i, i] \geq 14(k+2) \max_{\gamma \in \{1,2\}} n_\gamma &\geq 14(k+2) \sum_{j=1}^{k} M_\beta[h, j] \\
&\geq 14(k+2) \sum_{j=1}^{k} M_\beta[h, j] s_j = 14(k+2) T_\beta[h]
\end{aligned}
$$

Moreover, since $s_i \geq s_{k-1} \geq 1 - \frac{k-2}{k+1} = \frac{3}{k+1} > \frac{1}{k+1}$, we have $T_\beta[g] = M_\beta[g, i] s_i > \frac{M_\beta[e_i, i]}{k+1} > T_\beta[h]$ for each $\beta \in \{1, 2\}$. This gives $(T_1[g] - T_1[h])(T_2[g] - T_2[h]) > 0$, as desired.

- **Case 7:** $g = e_k$. By construction $M_\beta[e_k, j] > 0 \Rightarrow j = k$, and hence $T_\beta[g] = T_\beta[e_k] = M_\beta[e_k, k] s_k = 0$ for each $\beta \in \{1, 2\}$. If $h \in A \sqcup B \sqcup C \sqcup D$, we have $T_\beta[h] > 0$ for both $\beta \in \{1, 2\}$ by eqs. (A1) to (D2). If $h = e_i$ for some $i \in [k-1]$, on the other hand, $M_\beta[h, i] > 0$ by the last set of voters added to $\boldsymbol{\sigma}^{(1)}$ and $\boldsymbol{\sigma}^{(2)}$ (those that only rank the elements of $E$), so once again we have $T_\beta[h] = M_\beta[h, i] s_i > 0$. Hence, we have $(T_1[g] - T_1[h])(T_2[g] - T_2[h]) > 0$, as desired.

- **Case 8:** $g = e_i$, $h = e_j$ for some $i < j < k$. Fix some $\beta \in \{1, 2\}$. By construction, $\boldsymbol{\sigma}^{(\beta)}$ contains two types of rankings: (a) those that contain one elements in $A \sqcup B \sqcup C \sqcup D$ and all remaining elements are those in $E$, of which there are $n_\beta = \sum_{f \in A \sqcup B \sqcup C \sqcup D} \sum_{j=1}^{k} M_\beta[f, j]$ many and (b) those that rank only and all elements in $E$, of which there are say $y_\beta$ many, with $y_\beta \geq \max_{\gamma \in \{1,2\}} 14(k+2)n_\gamma \geq 14(k+2)n_\beta$. Hence, for any $j' \in [k-1]$, we have $n_\beta + y_\beta > M[e_{j'}, j'] \geq y_\beta$, where the first inequality is strict since $M_\beta[f, j'] > 0$ for some $f \in A \sqcup B \sqcup C \sqcup D$ for all $j' \in [k-1]$ (see, for example, the definitions for $M_\beta[a_{j'}, j'+1]$ and $M_\beta[b_{j'}, j']$ for each $j' \in [t]$), so there is at least some voters in the group of $n_\beta$ that do not rank $e_{j'}$. This implies

$$
\begin{aligned}
T_\beta[g] - T_\beta[h] = T_\beta[e_i] - T_\beta[e_j] &= M_\beta[e_i, i]s_i - M_\beta[e_j, j]s_j \\
&> y_\beta s_i - (n_\beta + y_\beta)s_j = y_\beta(s_i - s_j) - n_\beta s_j \\
&\geq y_\beta(s_i - s_{i+1}) - n_\beta = y_\beta \delta_i - n_\beta \\
&\geq 14(k+2)n_\beta \cdot \frac{\varepsilon}{2} - n_\beta = n_\beta - n_\beta = 0
\end{aligned}
$$

Hence, we have $(T_1[g] - T_1[h])(T_2[g] - T_2[h]) > 0$, as desired.

We have shown that $(T_1[g] - T_1[h])(T_2[g] - T_2[h]) > 0$ for all $g, h \in A$, proving that $s$ indeed gives full agreement between $\boldsymbol{\sigma}^{(1)}$ and $\boldsymbol{\sigma}^{(2)}$.

$\square$

## C  EXPERIMENT DETAILS

This appendix provides additional details on experiments discussed in Section 7.

Several components of our experiments are held constant across all settings. Below we describe these components and provide some general information about our experimental setting. In subsequent subsections we provide additional detail and discussion around each of our experiments and definitions of each voting rule used throughout this paper.

- *Number of Splits*: With two exceptions, all experiments report the average disagreement resulting from each rule over 10 random splits of voters. The exceptions are: When evaluating axioms (Appendix C.9, Figure 11) we report the average over 50 splits, and when evaluating score data (Section 7, Table 1) we report the average over 1000 splits. These values were decided based on computational and time constraints.

- *Splitting Procedure*: When creating two groups of voters we assign each voter to one group or the other uniformly at random, as detailed in Box 1. In the case that no voters are assigned to one group, we assign that group a single weak ranking in which all alternatives are tied (this effectively gives a weight of zero to all such splits).

- *Weighting Partial Rankings*: In settings where voters submit partial rankings, we weigh each alternative based on the number of times it appears on each side of a split, as discussed in Section 4. This serves to correct for mismatches in the amount of information known about each alternative (*i.e.*, the number of times each alternative is ranked) between the two sides of the split. Consider a split of a fixed profile $\boldsymbol{\sigma}$ into $(\boldsymbol{\sigma}^{(1)}, \boldsymbol{\sigma}^{(2)})$ via the process in Box 1. For any alternative $a \in A$, let $m_a$ denote the minimum number of times $a$ is included in rankings in $\boldsymbol{\sigma}^{(1)}$ or $\boldsymbol{\sigma}^{(2)}$, and let $t_a$ refer to the total number of times $a$ is included in a ranking across the entire profile $\boldsymbol{\sigma}$. Then the weight of $a$ is $w_a = \frac{\gamma^{m_a} - 1}{\gamma^{t_a/2} - 1}$. This ensures that the largest weights are assigned to alternatives that are evenly split across the split ($m_a = t_a/2$), whereas any alternative that does not appear on one side of the split ($m_a = 0$) gets zero weight, as it is unreasonable to expect this alternative to be placed consistently across the split by any SWF. All our experiments use a constant factor of $\gamma = 2$. Weights are used as a scaling factor in computing the Kendall-Tau distance (as in Kumar & Vassilvitskii 2010); when alternatives $a$ and $b$ are in opposite positions across two rankings then they contribute $w_a w_b$ to the weighted Kendall-Tau distance (as opposed to the unweighted setting where any misordered pair adds 1 to the distance).

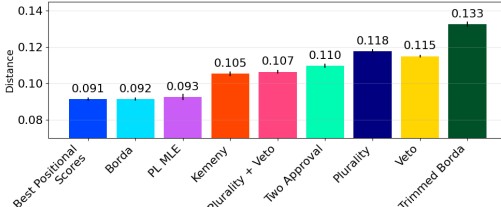 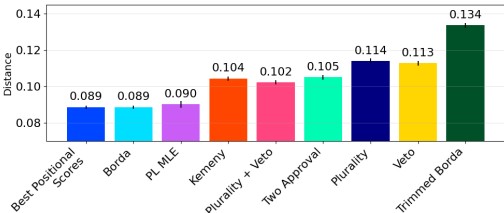

(a) Distance between splits of partial rankings for each SWF on reviews of ALMA Cycle 10 project proposals.

(b) Distance between splits of partial rankings for each SWF on reviews of ALMA Cycle 11 project proposals.

Figure 4: Split distance and standard error of several rules on partial rankings over project proposals for the Atacama Large Millimeter Array (ALMA).

- *Computational Power*: All experiments are performed locally on a 2022 M2 Macbook Air with 16 GB of memory. Each individual experiment shown in the paper took between 0.5 and 3 hours to complete.

## C.1 GROUND TRUTH AND DISAGREEMENT

In our first experiment, we evaluate the relationship between the rankings produced by SWFs and the ground truth ranking of a preference distribution.

Both Mallows and Plackett-Luce (PL) preference models naturally correspond to an "ideal" ranking. These distributions also have known maximum likelihood estimators: Mallow's MLE is the Kemeny function while we estimate the MLE of PL preferences using the `choix` Python library[6].

In our experiment, for each distribution we generate 50 elections with 100 voters and 100 alternatives. We set $\phi = 0.4$ for Mallow's preferences and $\alpha_i = e^{0.5(m-i)}$ for Plackett-Luce. Each voter provides a ranking over 10 alternatives, sampled from the relevant distribution, such that each alternative is ranked by 10 voters. This type of partial ranking aligns with a paper reviewing framework where conference organizers may wish to ensure that each paper is reviewed by a certain number of reviewers while each reviewer receives a certain number of assignments.

For each generated election, we do the following for several SWFs: Assign each voter into one of two groups, chosen uniformly at random for each voter. Each of these groups is then used as a complete profile to generate a weak ranking according to the SWF. We measure the $KT$ distance between these two rankings. We report the mean over these 10 splits as the "split distance." We also calculate the $KT$ distance from the ranking generated by the SWF applied to all voters to the ground truth ranking (the "ground truth distance").

We plot each pair of distances for each SWF. In Figure 1 the MLE of each noise model tends to minimize each distance. In general, there is a very strong relationship between the two distances. As the split distance increases, so too does the ground truth distance.

In this experiment the SWFs we evaluated were Kemeny, Plackett-Luce MLE (as implemented by the `choix` library), Borda Min-Max, and Optimized Positional Scores (see subsection C.10) in addition to positional scoring rules with the following vectors:

- Plurality $(1, 0, 0, \ldots, 0)$

- Plurality + Veto $(1, 0.5, 0.5, \ldots, 0.5, 0)$

- Veto $(1, 1, \ldots, 1, 0)$

- Two-Approval $(1, 1, 0, \ldots, 0)$

- Borda $\left(\frac{m-1}{m-1}, \frac{m-2}{m-1}, \ldots, \frac{m-m}{m-1}\right)$

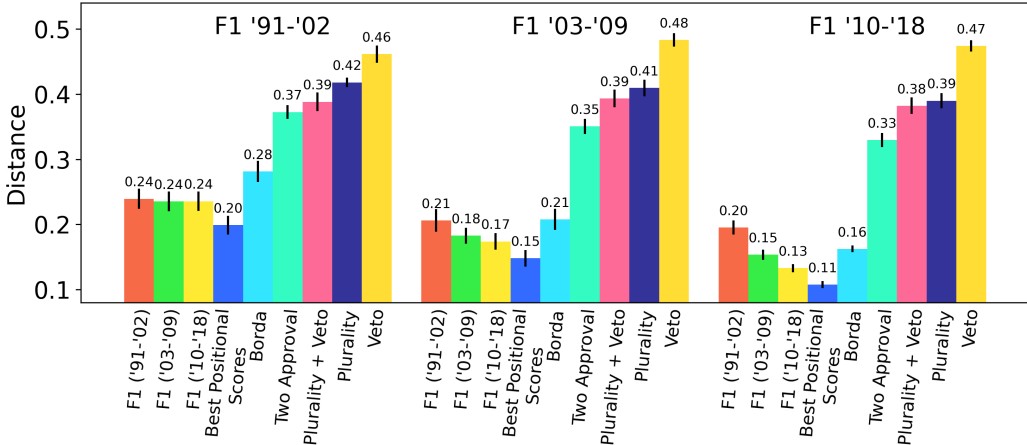

Figure 5: Distance between splits for SWFs aggregating rankings of drivers in F1 races. Each rule is evaluated on each period of races. Newer F1 rules provide lower distance on all race periods.

## C.2 ATACAMA LARGE MILLIMETER ARRAY PROJECT RANKING DATA

While in Table 1 we consider score data generated in a peer review process, peer review data can also take the form of rankings. Here we consider anonymized rankings provided by evaluators of projects proposed for the Atacama Large Millimeter Array (ALMA) (Meyer et al., 2022), which is the largest ground-based radio telescope on earth.

We conduct our experiments on two sets of proposals, from Cycle 10 and Cycle 11. In this review process, each proposer is asked to review. Cycle 10 contains 1635 proposals and 1635 reviewers, while Cycle 11 contains 1729 proposals and 1729 reviewers. In each cycle, every reviewer is assigned 10 proposals to review, and every proposal is assigned to 10 reviewers. Each reviewer provides a strict ranking of the 10 proposals assigned to them.

We run $AbC$ on the rankings provided by the reviewers. The mean Kendall-Tau distances over 10 splits for several SWFs are displayed in Figure 4. We see a pattern consistent with the results of Figure 1: PL MLE, Borda, and optimized positional scores result in very similar distances, suggesting a natural lower bound on the Kendall-Tau distance based on the data and the noise induced by generating splits. Kemeny rule,[7] on the other hand, performs notably worse than this lower bound. Our experiment also finds that better consistency is achieved by positional scoring rules which provide more information about the profile (*i.e.*, Two Approval and Plurality + Veto provide two "bits" of information, compared to one bit from Plurality and Veto, whereas Borda and the optimized positional scores provide significantly more information).

Importantly, our method can be used to evaluate proposed changes to current practices. For example, Meyer et al. (2022) consider removing the minimum & maximum score of each alternative before computing the Borda score, termed 'Trimmed Borda', but our experiment suggests this method increases disagreement on their dataset compared to Borda count, significantly more than it did on data generated from known noise models (Figure 1).

## C.3 FORMULA ONE DATA

The F1 portion of Figure 2 is generated using data about races found on Preflib Mattei & Walsh (2013); specifically, we use the complete form of dataset ID 00053. This contains one profile for each Formula One season with a preference order corresponding to each individual race in the season. Each preference order lists the drivers that competed in *every* race in that season, ordered by the position in which they finished that race.

---

[6]https://choix.lum.li/en/latest/index.html

[7]Note that, due to computational limits, in this setting we use the best ranking found by the Kemeny function (Appendix C.10.1) within a time limit of 15 minutes per split.

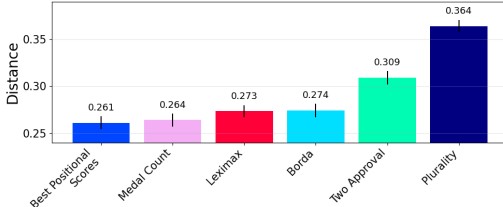 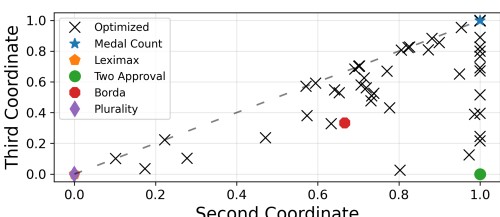

(a) Average distance and standard error between splits for SWFs over rankings induced by Olympic medals. Optimization of positional scores is only occasionally able to improve upon the ranking generated by giving one point to each country for each medal they win, regardless of medal type.

(b) Distribution of Best Positional Score vectors. Each × marker corresponds to the optimized score vector for a single Olympic games. Score vectors consist of three values (corresponding to Gold, Silver, Bronze winners) and are normalized so that their first coordinate is 1. The vector for Leximax is $(1, 10^{-3}, 10^{-6})$.

Figure 6: KT distance of various scoring rules (left)  and analysis of our optimization results (right) for Olympic Medal data.

We show in Figure 5 the $KT$ distance for all rules divided by racing period. While the rules with highest distance (Two-Approval, Plurality + Veto, Plurality, Veto) stay quite consistent, all other rules provider lower $KT$ distances on more recent race periods. Notably, the F1 rules themselves become much more consistent over time.

## C.4    OLYMPIC MEDAL DATA

To evaluate rules on Olympic medal wins we use a Kaggle dataset providing details of all Olympic results (Summer and Winter) between 1896 and 2016 (Griffin, 2018). From this we extract the winning countries of each medal for each event. We convert each event into a partial ranking by assigning to first place all countries receiving gold medals, to second place all countries receiving silver medals, and to third place all countries receiving bronze medals. Countries that did not win medals or did not compete in an event are not included in a ranking.

In the large majority of cases this results in a partial order consisting of exactly three countries, each in a different rank. In rare cases a position might be empty or have multiple winners (e.g. in 1992 Canada and USA won Gold in women's solo synchronized swimming, no Silver was awarded, and Bronze was won by Japan), or a country might occur multiple times (e.g. in 2008 Jamaica won Gold and two Silver medals in the women's 100 metre; no Bronze was awarded). In these exceptional cases we do nothing different and award that country points for each of the positions that it occupies.

This results in one "election" for each year in which the Olympics occurred where preference orders correspond to partial rankings induced by medal wins. To these rankings we apply the $AbC$ framework as we have in all other experiments: We generate splits by randomly placing the profile induced by each event into one split or the other, then find the distance between splits and report the average over many sets of splits. Here we add two new rules:

- **Medal Count:** Each medal is treated equivalently. Each country receives a point each time it appears in a preference order, regardless of rank.
- **Leximax:** Rank countries by the number of Gold medals they receive, breaking ties by counting Silver medals, breaking remaining ties by counting Bronze medals. In practice we implement this as the score vector $(1000000, 1000, 1)$.

Results of this analysis are displayed in Figure 6a. We see that simply counting the total number of medals each country receives results in much lower disagreement than standard voting rules, only very rarely do optimized scores improve upon the Medal Count rule.

Additionally, Figure 6b shows the resulting scoring vectors from our optimization ("Best Positional Scores"). Each positional scoring vector (with the points awarded for receiving Gold, Silver, Bronze respectively) is normalized to have a maximum value of 1 in the first coordinate (the number of points for each gold medal). The second and third coordinates are placed on the figure to map the

resulting score vector values. While optimized scoring vectors vary they consistently award a non-zero number of points for all medals, and the second coordinate is often approximately equal to either the first coordinate (i.e., a vector equal to $(1, 1, y)$) or the third coordinate (i.e., $(1, y, y)$). The Medal Count rule is the special case where both of these are true and all medals are worth an equal number of points.

## C.5 CITY ELECTION DATA

In Figure 2a we show results of several rules on Preflib data from several real-world political elections. Each of these elections used Instant Runoff Voting (IRV) to compute the empirical winner. Given a profile, IRV iteratively eliminates the alternative with the least plurality score (*i.e.*, with the least number of voters ranking them top) by removing them from the profile, until a single alternative remains. As an SWF, we interpret IRV as the rule outputting the alternatives in the reverse of this elimination order (the alternative eliminated first is ranked bottom, and so on). We provide details on each election shown in Table 2. The data from Preflib can be freely distributed and used under the GPL-3.0 license.

Initially we collected all elections included in Preflib Elections 5, 16, 17 ,18, 19, 20, 21, and 22. However, in Figure 2a we have filtered out two types of election from our starting data data in order to make the plot more readable:

- Any elections where all rules had a split distance of $0$. While this is a random event, we found that this consistently included a set of 10 elections.
- Any election with more than $100$ candidates. This excluded two specific elections that occurred in 2009 in the city of Minneapolis with 379 and 477 alternatives. These two elections have exceptionally high split distances and were removed as outliers due to their split distance and unusual number of candidates.

Table 3 lists rules in increasing order of $KT$ distance and shows how often each ordering occurred when $AbC$ was run on the 25 elections. As seen there, only two out of the 25 elections result in the same ranking of the rules in terms of their consistency when run on the election. This confirms our intuition from Section 1 that different rules are appropriate for different settings.

## C.6 MOVIELENS DATA

The MovieLens dataset (Harper & Konstan, 2015) is used widely to evaluate recommender systems. The data consists of 32,000,204 ratings (scores from 0.5 to 5 in increments of 0.5) of 87,585 movies by 200,948 users. We are able to apply $AbC$ to demonstrate how it might serve to generate ordered recommendation balancing quality and consistency. We take a subsample of the dataset to reach a tractable amount of data for use by our optimization method. Specifically, we filter the data to include only rankings from the 100 most prolific users, and filter that to include only the 100 most rated (across the entire dataset) movies. This results in 9401 ratings across the 100 movies. We convert the scores from each user into an ordinal ranking in order to evaluate the SWFs we have used throughout (although one could also use $AbC$ to pick among functions for aggregating cardinal evaluations, as demonstrated in Table 1 for peer review data).

Distance between splits on this data is shown in Figure 7. As is common in several of our experiments, Borda count outperform other non-optimized score vectors. In Table 4 we show the top and bottom movies in the ranking produced by the Best Positional Scores rule. The most highly ranked movies (of the 100 movies receiving the most rankings) consist of extremely well-known "classics" while the movies ranked at the bottom of the list are also well-known but may have less wide appeal (Ace Ventura and The Mask are primarily targeted at children) or were quite contentious at time of release (Star Wars: Episode I).

## C.7 ALTERNATIVE DISTANCE FUNCTIONS

As mentioned in Section 4, the $AbC$ framework can also be used in conjunction with distance functions other than the (weighted) Kendall-Tau distance. This can be especially for settings where the goal is not necessarily to rank all items but, instead, to select a set of winners or satisfy other criteria.

| Plot Index | Preflib Election ID | Election Name | # Voters | # Candidates |
|---|---|---|---|---|
| 0 | 5 | City (2009 Burlington Mayoral Election) | 8980 | 6 |
| 1 | 16 | City (Aspen City Council 2009) | 2477 | 11 |
| 2 | 16 | City (Aspen Mayor 2009) | 2527 | 5 |
| 3 | 17 | City (2010 Berkeley City Council - District 7) | 4173 | 4 |
| 4 | 18 | City (2009 Minneapolis Board of Estimate and Taxation Election - No Write In) | 32086 | 7 |
| 5 | 18 | City (2009 Minneapolis Park and Recreation Commissioner At-Large Election - No Write In) | 36655 | 9 |
| 6 | 19 | City (2010 Oakland Mayor) | 119256 | 11 |
| 7 | 19 | City (2010 Oakland City Council - District 4) | 20981 | 8 |
| 8 | 19 | City (2012 Oakland City Council - District 3) | 22079 | 7 |
| 9 | 19 | City (M2012 Oakland City Council - District 1) | 28660 | 8 |
| 10 | 20 | City (2008 Pierce County Assessor - Treasurer) | 262312 | 7 |
| 11 | 21 | City (2011 San Francisco Mayor) | 194530 | 25 |
| 12 | 21 | City (2011 San Francisco Sheriff) | 183192 | 5 |
| 13 | 21 | City (2012 San Francisco Board of Supervisors - District 5) | 35183 | 9 |
| 14 | 21 | City (San Francisco Board of Supervisors - District 7) | 31437 | 10 |
| 15 | 21 | City (2010 San Francisco Board of Supervisors - District 2) | 24109 | 7 |
| 16 | 21 | City (2008 San Francisco Board of Supervisors - District 9) | 26634 | 8 |
| 17 | 21 | City (2008 San Francisco - Board of Supervisors District 3) | 27310 | 10 |
| 18 | 21 | City (2008 San Francisco Board of Supervisors - District 1) | 28777 | 10 |
| 19 | 21 | City (2010 San Francisco Board of Supervisors - District 10) | 18001 | 22 |
| 20 | 21 | City (2008 San Francisco Board of Supervisors - District 11) | 24717 | 10 |
| 21 | 21 | City (2010 San Francisco Board of Supervisors - District 6) | 21188 | 15 |
| 22 | 22 | City (2010 San Leandro Mayor) | 22407 | 7 |
| 23 | 22 | City (2012 San Leandro City Council - District 4) | 23236 | 5 |
| 24 | 22 | City (2012 San Leandro City Council - District 2) | 25355 | 4 |

Table 2: Details on the election represented at each index in Figure 2a.

### C.7.1 JACCARD DISTANCE

Project proposals submitted to the Atacama Large Millimeter Array (ALMA) (Meyer et al., 2022) are ranked by several reviewers then either selected, or not selected.. Here, the goal is not necessarily to generate an aggregate ranking over all proposals, but to pick $k$ proposals to be funded, for some

| Count | Rule Order |
|---|---|
| 2 | Two Approval ≻ Veto ≻ Plurality ≻ IRV ≻ Borda ≻ Plurality + Veto |
| 1 | IRV ≻ Plurality + Veto ≻ Borda ≻ Veto ≻ Plurality ≻ Two Approval |
| 1 | Plurality + Veto ≻ IRV ≻ Veto ≻ Plurality ≻ Borda ≻ Two Approval |
| 1 | IRV ≻ Two Approval ≻ Plurality + Veto ≻ Plurality ≻ Veto ≻ Borda |
| 1 | Borda ≻ Plurality + Veto ≻ Veto ≻ Plurality ≻ Two Approval ≻ IRV |
| 1 | Plurality + Veto ≻ Plurality ≻ IRV ≻ Borda ≻ Veto ≻ Two Approval |
| 1 | Veto ≻ Plurality + Veto ≻ Two Approval ≻ Plurality ≻ IRV ≻ Borda |
| 1 | IRV ≻ Two Approval ≻ Plurality + Veto ≻ Plurality ≻ Borda ≻ Veto |
| 1 | Two Approval ≻ Borda ≻ Plurality ≻ Plurality + Veto ≻ IRV ≻ Veto |
| 1 | IRV ≻ Plurality + Veto ≻ Plurality ≻ Veto ≻ Borda ≻ Two Approval |
| 1 | IRV ≻ Two Approval ≻ Borda ≻ Plurality + Veto ≻ Veto ≻ Plurality |
| 1 | Borda ≻ Veto ≻ Two Approval ≻ Plurality + Veto ≻ IRV ≻ Plurality |
| 1 | Plurality + Veto ≻ Two Approval ≻ Veto ≻ Borda ≻ IRV ≻ Plurality |
| 1 | Two Approval ≻ Borda ≻ IRV ≻ Plurality + Veto ≻ Plurality ≻ Veto |
| 1 | Two Approval ≻ Veto ≻ Plurality + Veto ≻ IRV ≻ Borda ≻ Plurality |
| 1 | Borda ≻ Two Approval ≻ Veto ≻ IRV ≻ Plurality ≻ Plurality + Veto |
| 1 | Plurality ≻ Plurality + Veto ≻ Two Approval ≻ IRV ≻ Borda ≻ Veto |
| 1 | Plurality ≻ IRV ≻ Plurality + Veto ≻ Two Approval ≻ Borda ≻ Veto |
| 1 | Borda ≻ Veto ≻ IRV ≻ Plurality ≻ Plurality + Veto ≻ Two Approval |
| 1 | Two Approval ≻ Plurality + Veto ≻ Veto ≻ IRV ≻ Borda ≻ Plurality |
| 1 | Two Approval ≻ Veto ≻ Plurality + Veto ≻ Borda ≻ Plurality ≻ IRV |
| 1 | Borda ≻ Veto ≻ Two Approval ≻ IRV ≻ Plurality + Veto ≻ Plurality |
| 1 | Borda ≻ Veto ≻ Plurality ≻ Two Approval ≻ Plurality + Veto ≻ IRV |
| 1 | Plurality ≻ Plurality + Veto ≻ Borda ≻ Veto ≻ Two Approval ≻ IRV |

Table 3: The frequency of each rule ordering resulting from running $AbC$ across 25 political elections. Rules are listed in increasing order of $KT$ distance.

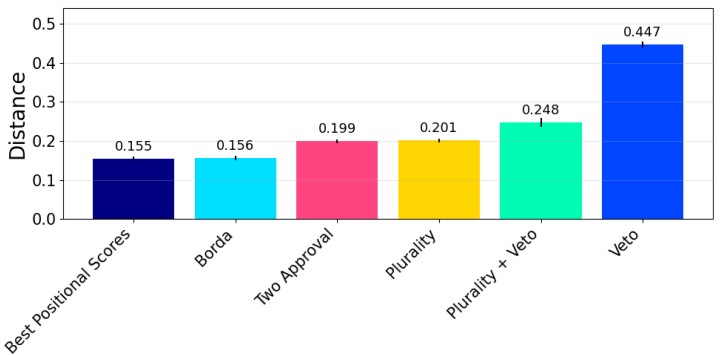

Figure 7: KT distance between splits for SWFs aggregating ratings of movies.

| Rank | Movie |
|------|-------|
| 1 | The Godfather (1972) |
| 2 | Pulp Fiction (1994) |
| 3 | Indiana Jones and the Raiders of the Lost Ark (1981) |
| 4 | Schindler's List (1993) |
| 5 | Star Wars: Episode V - The Empire Strikes Back (1980) |
| | |
| $m-4$ | Pretty Woman (1990) |
| $m-3$ | Mrs. Doubtfire (1993) |
| $m-2$ | Star Wars: Episode I - The Phantom Menace (1999) |
| $m-1$ | The Mask (1994) |
| $m$ | Ace Ventura: Pet Detective (1994) |

Table 4: Top and bottom 5 movies appearing in the ranking produced by the Best Positional Score vector.

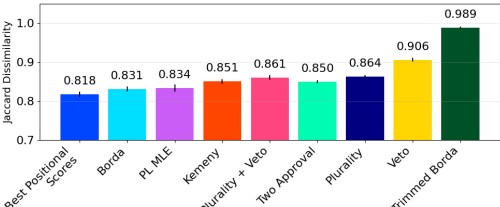

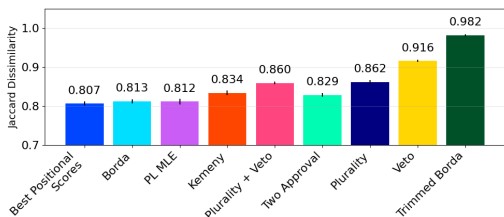

(a) Jaccard dissimilarity between sets of winners for splits of partial rankings of ALMA Cycle 10 project proposals.

(b) Jaccard dissimilarity between sets of winners for splits of partial rankings of ALMA Cycle 11 project proposals.

Figure 8: Jaccard dissimilarity and standard error of several rules on sets of winners using partial rankings over project proposals for the Atacama Large Millimeter Array (ALMA). Winners of a rule are the first 240 proposals in the ranking output by that rule.

$k < m$. It may thus be more useful to study the top $k$ proposals picked by each rule. Indeed, we explore this possibility by using the *Jaccard dissimilarity* (Jaccard, 1901) to measure the similarity of winning proposals on ALMA data.

*Jaccard index* measures the overlap of two sets; it is defined as the ratio of the size their intersection over the size of their union. *Jaccard dissimilarity* is then 1 minus the Jaccard index. For each voting rule that we evaluate, we select the 240 highest ranked proposals as the winning proposals (chosen to match the number of winning proposals chosen in ALMA Cycle 10).

In Figure 8 we see the Jaccard dissimilarity for several rules on proposal rankings from Cycle 10 and 11 of ALMA. The resulting disagreements have a similar relative order to the Kendall-Tau distance of the rules applied to the same data (Figure 4); in particular, we once again see that Trimmed Borda hurts consistency in comparison to Borda count.

### C.7.2 NORMALIZED DISCOUNTED CUMULATIVE GAIN

Normalized Discounted Cumulative Gain (NDCG) is a metric of ranking quality typically applied to search engine or information retrieval settings, but also used in recommender systems (Järvelin & Kekäläinen, 2002). NDCG evaluates a ranking based on the "relevance" of an item being ranked in a particular position, based on some ground truth ordering. As we do not have a ground truth against which we can evaluate a ranking, we measure the distance between a split of data by alternately treating the ranking resulting from each side of the split of the data as the optimal ranking (*i.e.*, we measure NDCG when one side of the split is the ground truth ranking and the other side is the ranking being evaluated. We then measure NDCG again when the roles of the two sides are switched. Finally, we compute the average between these two values). For $m$ total items being ranked we say that the relevance of the item being ranked in position $i$ is $m - i$. That is, for rankings

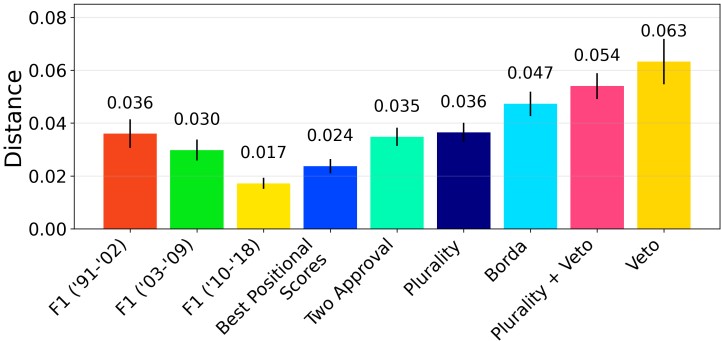

Figure 9: NDCG distance between splits for SWFs aggregating rankings of drivers in F1 races. F1 vectors are evaluated for only the years in which that rule was in use.

$(r_1, r_2)$ generated by a rule for a split of preference data we say that the distance between these rankings is equal to $1 - \frac{1}{2}(NDCG(r_1, r_2) + NDCG(r_2, r_1))$, where $NDCG(r, r_{opt})$ denotes the NDCG of $r$ when measured against the optimal ordering $r_{opt}$.

We calculate the NDCG score for a variety of voting rules on Formula One race data and present the results in Figure 9. While prior trends hold in that optimization of score vectors provide a significant improvement over most existing scoring rules (excepting the F1 (2010-2018) vector when applied only to races during those years), there is variation among the best performing non-optimized rules. While Borda scores have resulted in low distance on most other settings we find that Two Approval and Plurality are more effective at reducing NDCG distance, as both rules naturally focus on the ranking of alternatives that have been ranked top by most voters (and therefore will likely end near the top of the output ranking).

## C.8 DIFFERENT TIE WEIGHTS

As mentioned in Section 4, we explicitly add a term in our definition for $KT$ distance for ties in order to penalize indecisiveness (*e.g.*, a rule that always returns all alternatives tied). In all experiments above, ties in a ranking add to the $KT$ distance $\frac{1}{2}$ that of a strict disagreement, which is inspired by Kendall's Tau-b correlation coefficient (Kendall, 1945). Figure 10 reports the results of rerunning our experiment on F1 data (Figure 2b) while using different weights for the ties. The qualitative results are identical for the tie weights in $\{0.5, 0.75, 1\}$ in terms of the relative ordering of the F1 vectors, as well as in terms of the relative ordering of the named positional scoring vectors. For smaller tie weights (in $\{0, 0.5\}$) it is worth noting that the distances associated with Two-Approval, Plurality, Veto, and Plurality+Veto are significantly lower than they are for higher tie weights, whereas the distances associated with Borda count stay relatively constant across different tie weights. This makes intuitive sense, as the former four rules are more likely to result in final rankings with ties since each voter gives the same score to every alternative except one (for Plurality and Veto) or two (for Two-Approval and Plurality+Veto). In particular, all alternatives that have not won first place (resp. last place) in any race will be tied by Plurality (resp. by Veto) for every possible split of the profile. It is worth pointing out that different weights for the ties will result in different "Best Positional Scores" since our optimization minimizes the loss function that includes this weighted term for ties.

## C.9 AXIOM VIOLATIONS

To further explore the theoretical results of Section 5 we analyze experimentally how often certain axioms are violated across several distinct preference distributions. Results of this experiment are found in Figure 11. We measure the axiom violation rate, as defined by Caiata et al. (2025); an experimental measure of how often axioms are violated in practice on given preference distributions and voting rules. For this experiment we sample, for each distribution, 500 profiles with 100 voters that each provide a full ranking over $m \in 5, 10, 15, 20$ alternatives from the Impartial Culture, Urn (with $\alpha$ sampled from a Gamma distribution with shape $k = 0.8$ and scale $\theta = 1$ Boehmer

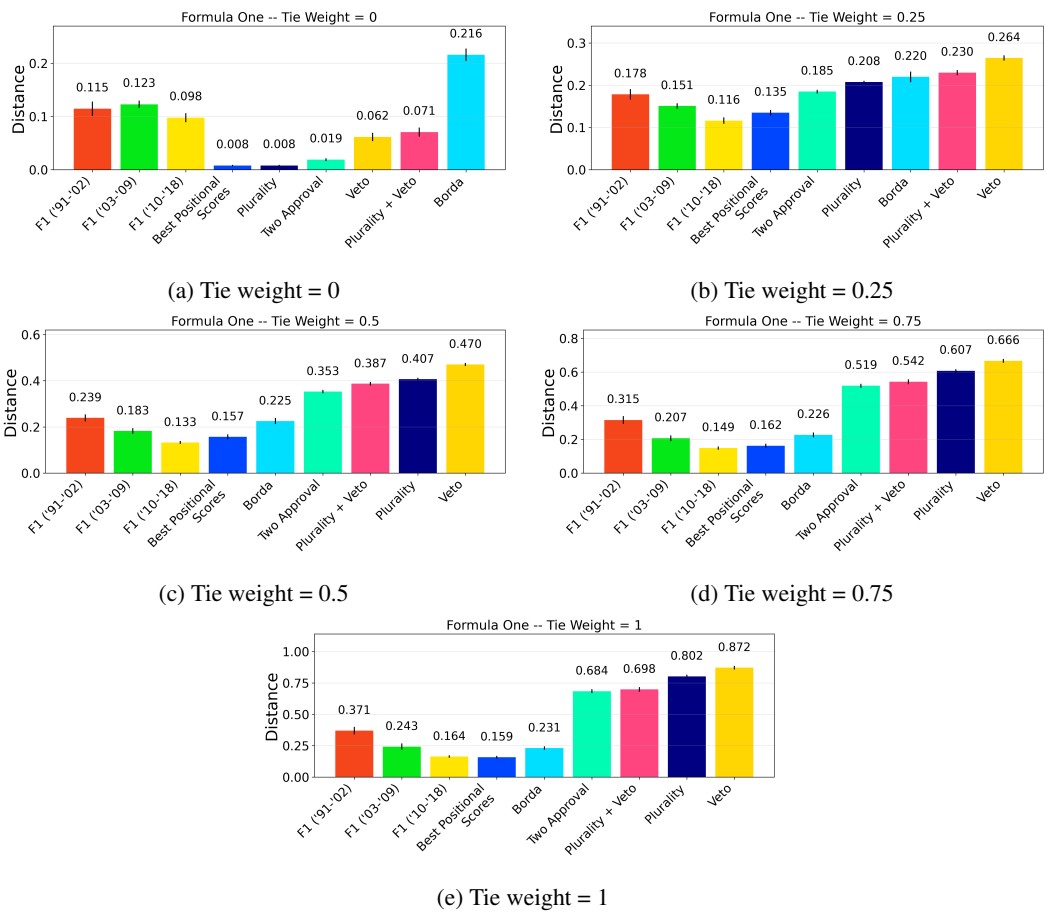

Figure 10: $KT$ distance between splits for SWFs aggregating rankings of drivers in F1 races, when ties are weighted by different coefficients (Definition 3). F1 rules are evaluated for only the years in which that rule was in use.

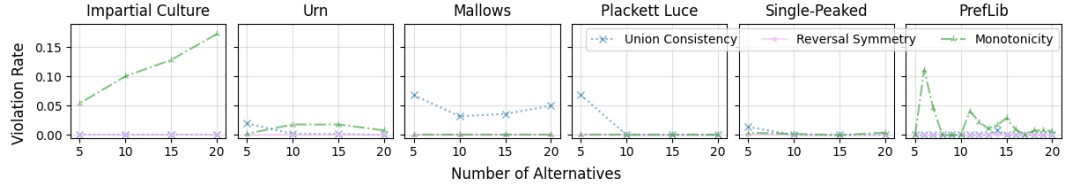

Figure 11: Violation rate of axioms across several preference distributions and real-world election data (rightmost). Axiom violations generally decrease as the number of alternatives increases.

et al. (2021)), Mallows ($\phi = 0.4$), Plackett-Luce ($\alpha_i = e^{0.5(m-i)}$), and Single-Peaked preference distributions Brandt et al. (2016). Additionally, we consider all elections with complete preferences from PrefLib with up to 1000 voters and $m$ alternatives for $5 \leq m \leq 20$ (a total of 1392 elections) Mattei & Walsh (2013). This real-world election data is compiled from a wide variety of sources with varying underlying preference distributions.

On each election we take 50 random splits and calculate the fraction of splits on which each axiom is violated by a Rule Picking Rule using several positional scoring rules.

As we evaluate the Reversal Symmetry axiom we are constrained to using positional scoring rules in our Rule Picking Rule. For all axioms, our RPR uses the following rules: Plurality, Plurality +

Veto, Veto, Two-Approval, and Borda count. The scoring vector associated with each of these rules is given in Appendix C.1.

We provide an example to illustrate how we measure axiom violations.

*Example.* Consider the monotonicity axiom. In a profile $\boldsymbol{\sigma}$ and a modification $\boldsymbol{\sigma}_a$ where some voters increase the rank they give to alternative $a$. Monotonicity is violated if the rank of $a$ under the RPR is lower in $\boldsymbol{\sigma}_a$ than in $\boldsymbol{\sigma}$.

We test for a violation of monotonicity in a profile $\boldsymbol{\sigma}$ by selecting the alternative $a$ ranked first in $\boldsymbol{\sigma}$ by the RPR, selecting some uniform random fraction of voters in $(0.2, 0.8)$ to increase their ranking of $a$. If the RPR on $\boldsymbol{\sigma}_a$ assigns a different rank to $a$ we say that the axiom has been violated. We report the fraction of instances tested in which the axiom is violated.

As shown in Proposition 10 (which easily extends to our implementation using Monte Carlo sampling), reversal symmetry is never violated and only checked as a test. Moreover, under all but one distribution, violations of union consistency and monotonicity are rare ($< 0.05$ of the time for $\geq 10$ alternatives) and generally decrease as the number of alternatives increases.[8]

## C.10 ADDITIONAL DEFINITIONS OF VOTING RULES

Through our experiments we use several rules which are not fully described. In this section we describe each rule found throughout our paper.

### C.10.1 KEMENY

Kemeny's rule is defined as the ranking which minimizes the sum of Kendall-Tau distances between the output ranking and each voter's individual ranking Kemeny (1959). We implement the Kemeny method code provided by Baharev et al. (2021) and the Gurobi optimization library (Baharev et al., 2021; Gurobi Optimization, LLC, 2024).

### C.10.2 TRIMMED BORDA

Calculate the number of voters ranking each alternative at each rank. For each alternative $a_i$, remove one of the highest rankings which $a_i$ has received and remove one of the lowest rankings which $a_i$ has received. After all removals are complete, calculate Borda scores as normal (Meyer et al., 2022).

### C.10.3 BEST POSITIONAL SCORES (SIMULATED ANNEALING)

This method uses the `optimal-voting` package generate a positional scoring vector which minimizes $KT$ distance between splits on a given profile (Armstrong, 2025). This package uses simulated annealing to generate novel positional scoring rules which optimize a target function. In each of our experiments, we run optimization multiple times; starting once from the initial score vector of each other positional scoring rule used in the experiment. For instance, if we were comparing with Borda's rule and Plurality we run optimization twice, starting from $(m-1, m-2, ..., 0)$ (Borda), and $(1, 0, ..., 0)$ (Plurality). In all cases, we run annealing for 500 steps. At each step, `optimal-voting` updates one index in the state vector (*i.e.*, the positional score vector) by some amount sampled uniformly at random from $(0.05, 1)$. We restrict updates to those that result in the state vector being weakly monotonically decreasing (the value at each index is not higher than the previous value). Using the updated state vector as a positional scoring vector we calculate the mean $KT$ split distance over each split of voters. We use the same set of splits in annealing as we do when evaluating each other voting rule. The new state is accepted with a probability related to the magnitude of difference between the current and previous mean split distances, and the number of steps that have already occurred. A small magnitude of difference is more likely to be accepted earlier than later. States with lower mean split distance are always accepted.

---

[8]The one exception to this trend is monotonicity in the Impartial Culture (IC) distribution, which is not centered around a ground truth. One possible explanation for this is IC maximizing the probability of majority cycles (Tsetlin et al., 2003), under which monotonicity violations are more likely to occur.

### C.10.4 STOCHASTIC GRADIENT DESCENT ON POSITIONAL SCORES

In Figure 3 we use Stochastic Gradient Descent (SGD) to find a positional score vector which minimizes $KT$ distance between sampled splits on a given profile. This method is included largely for comparison with the above approach which uses Simulated Annealing to optimize score vectors for the same metric. While both methods are effective at finding optimized positional scoring vectors, we find Simulated Annealing achieves results of slightly better quality with significantly less compute time.

In this method we apply the standard SGD approach, summing the $KT$ distance over all pairs of profile splits to generate the loss of a profile. Recall that $\mathbf{t}_\rho^s[a]$ is the total score of alternative $a$ under score vector $s$ for profile $\rho$ (Definition 1).

We approximate the $KT$ distance between two rankings $r_1$, $r_2$ using the sigmoid function. Note that here $\sigma$ refers to the sigmoid function rather than a profile and $q$ is some constant scaling used to scale the sigmoid function:

$$KT_{SGD}(r_1, r_2) = \sum_{i \in A} \sum_{j \in A, j > i} w_i w_j \sigma(q(\mathbf{t}_{r_1}(i) - \mathbf{t}_{r_1}(j))(\mathbf{s}_{r_2}(i) - \mathbf{s}_{r_2}(j)))$$

As $q \to \infty$, the sigmoid approaches the step functions, and $KT_{SGD}$ approaches $KT$ (Definition 3). Rankings $r_1$ and $r_2$ are calculated from the score vector for each pair of splits, and $KT_{SGD}$ is calculated for each pair of rankings. This is then normalized by the sum over each product of weights $w_i w_j$, which is the maximum value of $KT_{SGD}$ achievable for that given split. The total loss is the normalized sum of $KT_{SGD}$ over all splits in a profile.

We use the SGD Optimizer provided by Pytorch (Paszke et al., 2019) to perform gradient calculations. We initialize the scaling factor $q = 1000000$ and the learning rate to $0.1$. Learning rate is divided by $2$ if the loss from one step to the next increases by more than $0.1$ and has a minimum value of $10^{-6}$.

### C.10.5 LEXIMAX

This rule is used exclusively in evaluating Olympic data. For a given profile, alternatives are ranked according to the number of gold medals they have received. To break ties, tied alternatives are ordered according to the number of silver medals they have received. To break subsequent ties, tied alternatives are ordered according to the number of bronze medals they have received. Note that we can instantiate Leximax as a positional scoring rule where each position is *much* larger than the subsequent position. In our Olympic experiments we use the (pre-normalization) vector $(1000000, 1000, 1)$. As there are never more than $1000$ opportunities for a single country to receive a medal of one type this is equivalent to Leximax.

### C.10.6 MEDAL COUNT

This rule is used exclusively in evaluating Olympic data. For a given profile, alternatives are ranked purely based on the number of medals they received with no regard for the type of medals.

Our Olympics data uses as input the partial rankings containing only countries winning medals in an event. In this case, the Medal Count rule is equivalent to the positional scoring rule with the vector $(1, 1, 1)$.

