# OpenReview forum: "Designing Rules to Pick a Rule: Aggregation by Consistency"
_ICLR.cc/2026/Conference — ICLR 2026 Poster_

### Official Review · Reviewer_FWni · 2025-10-29

**Soundness:** 4
**Presentation:** 3
**Contribution:** 3
**Rating:** 6
**Confidence:** 3

**Summary:**

In all types of domains, individuals disagree. This disagreement is fundamental. As a result, rank aggregation rules must make tradeoffs, picking between different axioms that contradict each other to provide the best output. However, each methods tradeoffs mean that they are acceptable in different situations, and the correct choice of method will vary across domain and task. As a result, the authors here propose AbC (Aggregation by Consistency), a rule-picking rule, which provides a principled way to pick between existing methods. The authors propose a method of splitting data and comparing the consistency of given methods across those splits, allowing for the choice of a method well-suited for any given problem. The method splits voters (the authors explain how this can be used with incomplete preferences), and runs each chosen candidate rule on the split. They measure the Kendall-Tau of each method on each split, and pick the rule with the minimum expected disagreement.

In order to show the utility of their method, the authors explain what axioms their method satisfies, provide a few examples, and conclude with a set of experiments where they demonstrate AbC on a synthetic dataset, peer review dataset, and a F1 dataset.

**Strengths:**

*The problem the authors address is a fairly foundational one. Any given ranking rule must make choices on what to prioritize. These choices cannot generalize perfectly. A principled way of making that choice is a strong contribution and one that has a widespread impacts in many domains(as the authors note)

*While I have not checked the author's math in detail, it appears to be well-formulated and correct. The appendix contains a great deal of rigorous proofs to back up the author's assertions.

*The authors do a strong job not only in demonstrating the strengths and axioms their rules fulfill, but explain why they cannot fulfill the others. I did not find myself questioning any of those choices.

*The authors are explicit about which axioms they satisfy( reversal symmetry, plurality-shuffling consistency, unconditional anonymity, and neutrality).

*As an overall comment, I was satisfied with the writing of the paper, which was well-written and seemed to me very considered throughout.

**Weaknesses:**

The piece that stood out to me were the figures in the paper. They did not add much clarity to me, and in many cases they added to my confusion. Understanding the content of one given figure involves inspecting the appendix. What's good and what isn't doesn't jump out, and the number of methods felt like clutter and made it difficult for me to really get a sense of how each method performed. The labels are small and hard to read, 2A in particular is hard to get a sense of.

**Questions:**

One thing I wanted to ask the authors was about the use of 1/2 in place of a tie. Did you try any other values there, or measure any outcomes? In many settings such as RLHF, preferences are sparse. It does not seem unreasonable that one would want to avoid penalizing ties, or perhaps even the opposite.

---

> ### Author Response · Authors · 2025-11-21
>
> Thank you so much for your thoughtful review! We appreciate the time and effort dedicated to providing insightful comments and valuable suggestions. Below, we address each point separately.
>
> _“While I have not checked the author's math in detail, it appears to be well-formulated and correct. The appendix contains a great deal of rigorous proofs to back up the author's assertions.”_
> ---------
> Thank you for your encouraging comments on the theoretical claims in our paper. In addition to the full proofs in the appendix, __we have now added proof sketches to the main body after each of our three theorems__ (lines 345-350; 396-401; 420-426) to provide intuition on the proof techniques that we employed, along with explicit pointers to which subsection of the appendix the full proof can be found in. All new content added to the main body has currently been marked in __blue__. We hope the sketches will further help with the accessibility of our theoretical results.
>
> _Regarding the figures in the paper_
> ---------
> Due to the page limit, we had to keep the captions of the figures relatively short in the original submission. __We have now extended the caption of each figure__ (all new text marked in __blue__) to clarify the content and the main takeaways (“what's good and what isn't”). Please let us know if any of the figures need more explanation, and we would be happy to further clarify. For figure 2(a), the main idea that we wish to convey (as explained in lines 486-493 of the main body)  is that when we run our method on real-life election data, the relative ordering of the rules in terms of consistency differs widely from election to election, thereby confirming our intuition from Section 1 that different rules are appropriate for different settings. This also justifies the advantages of using a rule picking rule, since $AbC$ would pick the rule with the minimum disagreement for each election, thereby reliably selecting a consistent ranking. In fact, as we show in Table 3 in the Appendix (newly added, marked in __blue__), every single election we look at except two has a unique ranking of the rules in terms of consistency, displaying how the relative consistency of a given rule can strongly depend on the setting.
>
> _“One thing I wanted to ask the authors was about the use of 1/2 in place of a tie. Did you try any other values there, or measure any outcomes? In many settings such as RLHF, preferences are sparse. It does not seem unreasonable that one would want to avoid penalizing ties, or perhaps even the opposite.”_
> ---------
> As explained in line 212, the decision to weigh ties by 1/2 is inspired by Kendall’s Tau-b coefficient. Additionally, __we have now added a new experiment to our paper where we rerun $AbC$ on the F1 data with different weights/penalties for the ties__ (Appendix D.8, also mentioned in lines 212-213 in the main body; new text marked in __blue__). The main observation is that as the penalty for the ties increase, so does the disagreement associated with rules such as plurality and veto, whereas the disagreement associated with Borda count stays relatively constant. This makes intuitive sense, as the former two rules are more likely to result in final rankings with ties since each voter gives the same score to every alternative except one. In particular, all alternatives that have not won first place (resp. last place) in any race will be tied by plurality (resp. by veto) for every possible split of the profile.
>
> We would also like to emphasize that __we are not penalizing ties in voters’ preferences, but rather ties in the final aggregate ranking__. So even in a setting such as RLHF, where voters’ preferences are sparse, there would be no penalty associated with ties as long as the final output ranking is not indifferent between alternatives. Indeed, in the partial ranking setting we discuss and implement (lines 259-268; 462-470; Appendix D.1), each voter is allowed to rank a small subset of all alternatives, and no assumption is made about their preferences on the remaining alternatives.

---

> > ### Comment · Reviewer_FWni · 2025-11-26
> > **Response to authors**
> >
> > I thank the authors for responding to my comments, and apologize for the delayed response. I am satisfied with their responses to my questions. I recognize the space constraints and I am happy to see the authors attempt to make changes as requested.
> >
> > I plan to leave my score at least as is; after I review the discussions with other reviewers I may choose to raise it futther.

---

### Official Review · Reviewer_iZ34 · 2025-10-30

**Soundness:** 4
**Presentation:** 4
**Contribution:** 3
**Rating:** 6
**Confidence:** 4

**Summary:**

This paper studies a Rules to Pick a Rule framework: given a ranking profile, what is the ``best'' social choice function to make an aggregation? This paper proposes a surprisingly simple yet effective method: split the profile into two halves u.a.r, run SCF candidates on both halves, and pick the SCF where two outcomes agrees the most (i.e. smallest Kendall-Tau distance between two aggregated rankings). The basic idea is that a voting rule should be consistent among ranking data drawn from the same distribution. Theoretically, this paper considers a series of axioms where their consistence RPR satisfies or those nobody can do better. Empirically, the show the Monte-Carlo approximation of their RPR presents pretty good outcome on datasets.

**Strengths:**

This paper is quite impressive to me. In the introduction, the paper exhibits a very strong connection with previous studies and place itself well among them. They also clearly justified how their key idea -- consistency -- is natural and intuitive in many interdiscinpline research topics. I don't check the proofs, but the theorems sounds reasonable at a high level. This paper give a reasonably significant conceptual contribution in rank aggregation with a suprisingly simple method, which I think is the most valuable part of this paper. Besides that, it has a thorough picture, from theory, to computation, and to emprical studies. The topic is also very related to ICLR.

**Weaknesses:**

I am not 100% percent convinced by the idea of "rules to pick a rule". For me, it sounds perfectly ok to interpret it as a another type of rank aggregation rules (as the paper said, the rule RPR induces), which brings more explainability on "we always align with (one of) the SCF with highest consistency".  Specifically, given the method is profile-based, the rule elected is used solely for this data. What is the specific reason to motivate and interpret as a "rule to pick a rule"?

Another concern is that, I feel a little bit inconsistent on the motivation of this paper and the approach it adopts. Consider two area of works. (1) Data driven rank aggregation like preference learning and RLHF. In these scenarios, rankings are all datas, and what rules to pick wholely relies on your goal (like maximizing consistency or minimizing other loss functions). (2) Traditional social choice. In this area, "rules to pick a rule" based on voting profiles is very problematic as a strong signal of manipulation. This paper adopts the scenarios in (1) yet adopt the methodology in (2). Given that the overall goal of this paper is to justify "consistency is a gold standard on picking SCF", the axiomatic approach, which sounds more like "consistency brings natural behaviors under certain hypothetical scenarios", is not a strong support to the paper. How would you answer to this incosistency and justify your axiomatic analysis?

**Questions:**

1. What is the specific reason to motivate and interpret as a "rule to pick a rule"? (Weakness 1)
2. How would you answer to this incosistency and justify your axiomatic analysis?  (Weakness 2)
3. Are there any references to the given consistency axioms? Specifically, why "reverse symmetry" sounds so important? At first glance it does not look like a fundamental axiom like anonymity or neutrality.
4. Is your AbC rule not welfare maximization? If so, do you have any comments on this?

I am open to discussion and happy to raise my score if my concerns are addressed.

---

> ### Author Response · Authors · 2025-11-21
> **Author response [1/3]**
>
> Thank you so much for your encouraging and detailed review! We appreciate the time and effort dedicated to providing insightful comments and valuable suggestions. Below we respond to each of your questions one by one.
>
> _What is the specific reason to motivate and interpret as a "rule to pick a rule"? (Weakness 1)_
> ---------
> We believe that interpreting $AbC$ as a rule picking rule rather than simply the rule it induces has several advantages:
> 1) Interpretability: In practice, $AbC$ picks among rules whose output is easy to interpret and verify (e.g., for positional scoring rules, it is easy to compute the total score of each alternative given the scoring vector). However, interpreting the output of the social welfare function (SWF) induced by $AbC$ (i.e. the rule that picks the most consistent rule and then immediately applies it to the data, as defined in lines 354-360) using the output ranking alone (without knowing which candidate rule was picked by $AbC$) is much more difficult, since the most consistent rule may change from profile to profile. In this sense, we believe outputting the specific candidate rule that provides the highest consistency (the formal output of $AbC$ as a rule picking rule) is an important intermediary step for interpretability, before this rule is used for aggregating the rankings.
> 2) Perhaps more importantly, rule picking rules do not just take the profile as an input, but also a set of candidate rules (Definition 2). As a result, the induced SWF will be different for different sets of candidate rules. The classic model of rules that map rankings to an output is not rich enough to capture this dependence. Having candidate rules as part of the input makes our model general enough to allow the implementor to list a set of acceptable rules, and then our method will pick among them. For example, as we see in Section 5.2 (Preserved Axioms), many important social choice axioms can be imposed on the rule induced by $AbC$ by restricting the candidate set to rules that satisfy them. So in a setting where, for example, Condorcet consistency is very important, the candidate rule set can be restricted to such rules. In other settings where we have reason to believe there is some ground truth ranking (e.g., lines 462-470), it might be more appropriate to restrict our candidate rules to MLEs of known noise models. Further, $AbC$ can be continually updated by designing and implementing new rules into the candidate rule set. Overall, we believe picking rules rather than directly outputting rankings has several advantages, as we also discuss in lines 065-072.
> 3) In general, $AbC$ gives a principled methodology to compare any two rules in terms of their consistency. This allows users to evaluate the impact of proposed modifications to existing methods by running $AbC$ on their own data. In some cases, the answer is that it is an improvement: both changes to F1 rules significantly reduced disagreement (Fig. 2b). In others, the proposed modification hurts consistency: ALMA telescope considers adding outlier rejection to Borda for peer review, but our experiment suggests this method (Trimmed Borda) increases disagreement on their dataset (lines 477-481, new text is marked in __blue__).

---

> > ### Author Response · Authors · 2025-11-21
> > **Author response [2/3]**
> >
> > _“How would you answer to this incosistency and justify your axiomatic analysis? (Weakness 2)”_
> > ---------
> > This is a great question. We would first like to point out that there has been nascent and growing interest in applying tools and axioms from social choice to data-driven aggregation settings like RLHF (e.g., [Conitzer et al., 2024; Dai & Fleisig, 2024; Ge et al., 2024; Mishra, 2023], among others). By contributing to this body of work, we hope that our paper paves the way for future applications of social choice to data-driven settings. In the other direction, we would like to argue that rule picking rules are also relevant for various settings in traditional social choice and can be implemented in a way that does not necessarily allow novel forms of manipulation. In particular, just like we must commit to a rule in traditional social choice, we can always commit to the specific _rule picking rule_ (in our case, $AbC$) and the set of candidate rules prior to the voting/evaluation process, which prevents a manipulator from arbitrarily picking a rule based on the collected rankings. Even if we will be using our Monte-Carlo sampling method (section 7) rather than the theoretical optimum, any seeds of randomness that will be used can be released prior to the election/evaluation to ensure full transparency. Indeed, there is a rich history of studying randomized rules in traditional social choice (see, for example, [Brandl et al., 2016]). In other settings, we may have to commit to a simple aggregation rule upfront before collecting the evaluations/votes, but we can do so based on past data, in which case the rule picking rule can once again be employed. For example, say you are the program chair of ICLR 2026, and you have partnered with researchers to run a peer review experiment on your conference. You are given access to only half of the reviews now, and you must pick a rule to aggregate them; later, the peer review researchers will release the rest of the reviews, checking the consistency between the accepted papers (similar experiments were run in NeurIPS 2014 and 2021). In online settings such as these, where you might have to pick a rule prior to seeing the entire profile, AbC as a rule picking rule can be extremely useful for maximizing consistency. Please let us know if you have any further questions about this or would like us to clarify further!
> >
> >
> > _“Are there any references to the given consistency axioms? Specifically, why "reverse symmetry" sounds so important? At first glance it does not look like a fundamental axiom like anonymity or neutrality.”_
> > ---------
> > As noted in line 284, our definition for reversal symmetry is inspired by the homonymous property for rank aggregation methods introduced by Saari [1994]. In that context, reversal symmetry dictates that if we reverse every voter’s ranking, then the output of the method should also be the reverse of its original output. Holliday and Pacuit [2023] adapt this axiom to social choice functions (which, given a profile, return a subset of alternatives); here, reversal symmetry requires that if an alternative is the unique winner of a profile, it should not be a winner of the reversed profile. Our axiom for RPRs is more demanding in the sense that we not only require the picked positional scoring rules to change, but also that they should be replaced exactly by their reverse rules. Union consistency is also inspired by an analogous axiom for voting rules (lines 327-328), simply named consistency [Young, 1975]. Plurality-shuffling-consistency, on the other hand, is a new axiom that relies on novel definitions (such as “shuffled profiles”) introduced by us specifically for the setting of rule picking rules.
> >
> > Anonymity and neutrality are indeed fundamental axioms that describe some minimum properties any “acceptable” rule should satisfy. Axioms like reversal symmetry and union consistency, on the other hand, are not “must”s for acceptability, but additional desirable properties that a “good” rule picking rule would satisfy (similar to axioms like independence of clones, monotonicity, etc. in single winner voting). In particular, the normative value for these axioms comes from encoding “sanity checks” on the rule picking rule to ensure it behaves desirably when we change the profile(s) in certain ways. For example, if a rule picking rule picks plurality for a specific profile, this means it has concluded that we must be using the top position of every voter’s ranking (perhaps because the signal is concentrated there, as is the case with shuffled profiles). Then if we reverse the profile, it is natural to expect the same rule picking rule to now pick veto, because whatever signal used to be in the top positions of the voters’ rankings is now in their bottom positions.

---

> ### Author Response · Authors · 2025-11-21
> **Author response [3/3]**
>
> _“Is your AbC rule not welfare maximization? If so, do you have any comments on this?”_
> ---------
> If the reviewer is asking whether AbC fits into the class of welfare-maximizing rule picking rules as defined in Definition 8, the answer is no, which is a natural corollary of our theorem 1: AbC satisfies plurality-shuffling-consistency, whereas every welfare-maximizing rule picking rule fails it. If the reviewer meant something else, please let us know and we would be happy to further discuss.
>
>
> __References:__
> - Florian Brandl, Felix Brandt, and Hans Georg Seedig. Consistent probabilistic social choice. Econometrica, 84(5):1839–1880, 2016.
> - Vincent Conitzer, Rachel Freedman, Jobst Heitzig, Wesley H. Holliday, Bob M. Jacobs, Nathan
> Lambert, Milan Mosse, Eric Pacuit, Stuart Russell, Hailey Schoelkopf, Emanuel Tewolde, and
> William S. Zwicker. Social choice should guide AI alignment in dealing with diverse human
> feedback. In Proceedings of the International Conference on Machine Learning (ICML), 2024.
> - Jessica Dai and Eve Fleisig. Mapping social choice theory to RLHF. arXiv preprint
> arXiv:2404.13038, 2024.
> - ​​Luise Ge, Daniel Halpern, Evi Micha, Ariel D. Procaccia, Itai Shapira, Yevgeniy Vorobeychik, and Junlin Wu. Axioms for AI alignment from human feedback, 2024.
> - Wesley H. Holliday and Eric Pacuit. Split cycle: a new condorcet-consistent voting method independent of clones and immune to spoilers. Public Choice, 197(1-2):1–62, August 2023.
> - Abhilash Mishra. AI alignment and social choice: Fundamental limitations and policy implications. arXiv preprint arXiv:2310.16048, 2023.
> - D.G. Saari. Geometry of Voting. Springer Berlin Heidelberg, 1994.
> - H. P. Young. Social choice scoring functions. SIAM Journal on Applied Mathematics, 28(4):824–838, 1975.

---

### Official Review · Reviewer_aygk · 2025-10-31

**Soundness:** 3
**Presentation:** 3
**Contribution:** 3
**Rating:** 6
**Confidence:** 3

**Summary:**

The paper introduces Rule Picking Rules (RPRs) : procedures that choose which aggregation rule to use for a given dataset; and proposes Aggregation by Consistency (AbC), which selects the rule whose outcomes are most stable across random evaluator splits.

It develops an axiomatic basis (e.g., reversal symmetry, shuffling consistency), proves that achieving perfect consistency over positional scoring rules is NP-complete, and provides a computationally practical Monte-Carlo implementation.

AbC (i) often rediscovers the model-MLE when the generative model matches the data, (ii) yields actionable guidance on rating aggregations, and (iii) extends to top-k selection via set-based distances.

**Strengths:**

The paper is well written and answers an important problem, especially motivated by the Neurips paper acceptance experiment.

The axiom satisfactions depict that the proposed method meets sanity checks.

The proofs in the appendix seem thorough.

The monte carlo estimation makes the implementation practical.

**Weaknesses:**

The paper focuses on Kendall's Tau as the rank comparision metric. Metrics like Cumulative Gain (CG), Discounted CG (DCG), and Normalized DCG are popular for comparing top heavy ranks. A discussion involving those metrics and how the algorithms fares to them would be good to have.

The empirical section is somewhat narrowly scoped. Although they show correlation between disagreement and distance to ground truth in the synthetic models, no end-to-end evaluation links “most consistent rule” to “best downstream decision quality.” For instance, in peer review: does the AbC-selected rule produce higher eventual citation impact or expert consensus?

Moreover, the real-world data are both from astronomy. It would be helpful to experiment on a diverse set of domains (e.g., recommendation rankings, RLHF preference data, crowd judgments - all readily available) to show generality.

**Questions:**

Did AbC ever select a rule whose behavior appears to be intermediate between known positional scoring rules (e.g., between Kemeny and Borda)? Did it ever select a rule not a combination of existing rules?

When optimizing over positional weights to minimize split-disagreement, do the resulting weights cluster around classical forms (e.g., linear, exponential), or is there a continuum?

Are there theoretical conditions under which AbC provably recovers Kemeny or PL-MLE — beyond the empirical confirmation on Mallows/PL data?

Is there any identifiable set of axioms that exactly describe the rules that minimize disagreement under random resampling? I.e can you simulate an uniqueness axiom?

---

> ### Author Response · Authors · 2025-11-21
> **Author response [1/2]**
>
> We would like to thank the reviewer for their thoughtful and detailed review. We appreciate the time and effort dedicated to providing insightful comments and valuable suggestions. Below we address each point one by one.
>
> _“The paper focuses on Kendall's Tau as the rank comparison metric. Metrics like Cumulative Gain (CG), Discounted CG (DCG), and Normalized DCG are popular for comparing top heavy ranks. A discussion involving those metrics and how the algorithms fares to them would be good to have.”_
> ---------
> While we have defined $AbC$ using the Kendall-Tau distance function for our axiomatic analysis in Section 5, the $AbC$ framework can indeed be used in conjunction with other distance functions, as mentioned in Section 4 (lines 202-203). In fact, in Appendix D.7 of our original submission, we had run additional experiments on the anonymized peer review data provided by the Atacama Large Millimeter Array (ALMA) telescope [Meyer et al., 2022] using Jaccard dissimilarity as a distance measure rather than Kendall-Tau. For ALMA, the goal is to pick a subset of proposals  (of fixed size for some $k<m$) to fund out of all alternatives, and the ranking among these “winners” is not important. Therefore, using a distance between subsets of alternatives like Jaccard dissimilarity instead of Kendall-Tau is a reasonable alternative. __We have now extended our discussion of these experiments in the main body, and added a new experiment we have implemented using the Normalized Discounted Cumulative Gain (NDCG) on the Formula 1 (F1) data.__ In F1, it might indeed make sense to prioritize disagreements about alternatives on top of the rankings above lower ranked ones (“top heavy”), so NDCG is a reasonable alternative to KT distance; we thank the reviewer for this valuable suggestion. The experiments using Jaccard dissimilarity and NDCG can be found in Appendix D.7 in the new draft we have uploaded, and they are now also discussed in the main body (lines 498-510). All new text is currently marked in __blue__. Overall, we see that the two alternative distance functions lead to qualitatively similar results to those using Kendall-Tau distance.
>
> _“For instance, in peer review: does the AbC-selected rule produce higher eventual citation impact or expert consensus?”_
> ---------
> Using eventual citation count as a measure of best downstream decision quality can be problematic due to our inability to measure counterfactuals: naturally the citation counts of the papers that ended up getting accepted were affected by this acceptance decision, so if we compare the rule that was (in real life) used for this decision with what papers $AbC$ would have accepted / ranked top, we would not be able to make a fair comparison. One potential trick around this challenge would be to restrict our set to the accepted papers, and compare how $AbC$ ranks them with their citation counts. However, it has also been shown in several peer review experiments that there is no correlation between the reviewer scores for an accepted paper and that paper’s eventual citation count [Cortes and Lawrence 2021, Ragone et al. 2013, Connolly et al. 2014], which makes it difficult to use citation count as a measure of quality for any function of the reviewer ratings/rankings. We believe it is due to challenges such as these that there has been no universally accepted evaluation metric adopted in peer review research. An important part of our argument in this paper is that consistency itself should be considered a metric of decision quality, in part due to the connections we draw in Section 2 and the axiomatic properties we prove in Section 5. In a way, our method maximizes expert consensus “by definition,” at least among the evaluators whose rankings we are aggregating. In any case, we agree that studying the relationship between consistency and other measures of decision quality is an important avenue of future research.
>
> _“Moreover, the real-world data are both from astronomy. It would be helpful to experiment on a diverse set of domains (e.g., recommendation rankings, RLHF preference data, crowd judgments - all readily available) to show generality.”_
> ---------
> We would like to clarify that in addition to the two peer review experiments for astronomy, our original submission included experiments run on real-life data from Formula 1 races, Olympics, and city elections with many voters. We believe that the generality of $AbC$ is one of its important strengths, and future research can implement it in all kinds of domains, which we find exciting. Among the suggestions provided by the reviewer, we found that an implementation on recommendation systems is the most straightforward, and __we have now added a new experiment running $AbC$ on the MovieLens dataset__, which we present in Appendix D.6 and discuss in lines 494-497 of the main body of the new draft (all new text is marked in __blue__). Please let us know if you would like to see any further additions.

---

> ### Author Response · Authors · 2025-11-21
> **Author response [2/2]**
>
> _“Did AbC ever select a rule whose behavior appears to be intermediate between known positional scoring rules (e.g., between Kemeny and Borda)? Did it ever select a rule not a combination of existing rules?”_
> ---------
> The only parameterized class of voting rules we have implemented in this paper is positional scoring rules (by running an optimization over them, see lines. 512-523). To this, we are also able to add any additional rules, such as MLEs of noise models, e.g. Kemeny and the MLE of Plackett–Luce (Fig. 1), which are not positional scoring rules. Since a rule picking rule gets the set of candidate rules as an input in addition to the profile (Definition 2), it is by definition not allowed to output a rule not in the candidate set (although this set can be infinitely large, as in the case of positional scoring rules). Please let us know if you would like us to further clarify.
>
> _“When optimizing over positional weights to minimize split-disagreement, do the resulting weights cluster around classical forms (e.g., linear, exponential), or is there a continuum?”_
> ---------
> While illustrating trends in the optimal positional scoring weights in general is difficult (since it is an $m$-dimensional vector), for the experiment we run on Olympics data (Appendix D.4), the vector only contains three numbers (the points for receiving a Gold, Silver, or Bronze medal), and the top position (Gold) is normalized to 1, so there are only 2 free variables, which makes illustration easier. Inspired by the reviewer’s question, __we have now added a new figure into the appendix showing the distribution of the optimized positional weights__ in this experiment (Figure 6(b)) in the appendix, where the two axes correspond to the scores assigned to second (Silver) and third (Bronze) positions. As seen in the figure and discussed in lines 1725-1732 (all new text marked in __blue__), we see that the second coordinate of the optimized vector is often approximately equal to either the first coordinate (i.e., a vector equal to $(1, 1, y)$) or the third coordinate (i.e., $(1, y, y)$), resulting in clusters of points forming two distinct lines. We would also like to recall that no point can end up in the upper left triangle of the figure, as we impose monotonicity of the positional scoring rule ($s_1 \geq s_2 \geq \ldots \geq s_m$).
>
> _“Are there theoretical conditions under which AbC provably recovers Kemeny or PL-MLE — beyond the empirical confirmation on Mallows/PL data?”_
> ---------
>
> This is a great question! While we do not have a definitive answer, we can point out that it is difficult to obtain such conditions without any assumptions on the candidate rule set too. For example, if the candidate rule set includes (in addition to Kemeny) a rule that outputs the reverse of the Kemeny ranking, this rule will always obtain the same consistency across splits as Kemeny itself (regardless of the size of the profile), so $AbC$ cannot recover Kemeny as long as unreasonable rules like this are included in the candidate set. We consider finding minimum axioms/conditions for the candidate rule set and the profile to theoretically recover the MLE to be a valuable future question.
>
>
> _Is there any identifiable set of axioms that exactly describe the rules that minimize disagreement under random resampling? I.e can you simulate an uniqueness axiom?_
> ---------
> While we give some impossibility results for rule picking rules in this paper (e.g., reversal symmetry + plurality-shuffling-consistency + union consistency in Theorem 1; reversal symmetry + preservation of monotonicity in Theorem 2), we do not currently have a set of axioms that fully characterize $AbC$. We hope our work paves the way for future research in studying axioms for rule picking rules and characterizations that result from them.
>
> __References:__
> - Randy Connolly, Janet Miller, and Rob Friedman. 2014. A longitudinal examination of SIGITE conference submission data, 2007-2012. In Proceedings of the 15th Annual Conference on Information technology education (SIGITE '14). Association for Computing Machinery, New York, NY, USA, 167–172.
> - Corinna Cortes and Neil D. Lawrence. Inconsistency in conference peer review: Revisiting the 2014 NeurIPS experiment, 2021.
> - Jennifer Donovan Meyer, Andrea Corvillon, John M Carpenter, Adele L Plunkett, Robert Kurowski, Alex Chalevin, Jakob Bruenker, D-C Kim, and Enrique Macias. Analysis of the ALMA cycle 8 distributed peer review process. arXiv preprint arXiv:2204.05390, 2022
> - A. Ragone, K. Mirylenka, F. Casati, et al. On peer review in computer science: analysis of its effectiveness and suggestions for improvement. Scientometrics 97, 317–356 (2013).

---

### Official Review · Reviewer_6kf7 · 2025-11-01

**Soundness:** 3
**Presentation:** 3
**Contribution:** 3
**Rating:** 8
**Confidence:** 3

**Summary:**

The paper tackles the problem of selecting the best rank aggregation method in the setting where different aggregation rules can lead to very different outcomes. To address this problem, the authors introduce the concept of a Rule Picking Rule (RPR), a meta-framework for deciding which aggregation method to use for a given dataset without assuming any specific generative model.  Instead of relying on fixed axioms or assumed noise models, the authors  provide a data-driven approach for picking aggregation rules. They propose a new RPR, called "AbC", which selects the aggregation method in such a way to maximize consistency, that is to say, producing stable results if the data collection were repeated. The authors show that AbC satisfies certain consistency-related axioms that some natural RPRs fail to meet.  For two axioms it cannot satisfy, the authors prove impossibility results, showing no rule can satisfy them simultaneously. Additionally they show that the problem of finding a perfectly consistent rule is NP-complete. Despite this hardness result, the authors have implemented a sampling-based algorithm that seems to perform reasonably well, in the sense that it outperforms some commonly used aggregation methods. The implementation also provide interpretable insights into when and why specific rules work best. It is stated that an implementation of AbC was awarded in a recent competition
at a top-tier AI conference.

**Strengths:**

The main strength of the paper is that the proposed RPR has been implemented and evaluated in practice, and obtained one of the top scores in a relevant AI competition.

**Weaknesses:**

The theoretical part is dense and most proofs are relegated to the appendix, which is quite long (22 pages). Therefore, verifying the correctness of the claims is a bit tricky. I did not verify the correctness of the proofs in the appendix.

**Questions:**

It is stated that an implementation of AbC was awarded in a recent competition at a top-tier AI conference.

1) What place/score did the mentioned implementation get? 1st place, 2nd place, 3rd place?
2) How many competitors were there in the competition?

Without more details it is difficult to judge the strength of the implementation with respect to competing approaches.

---

> ### Author Response · Authors · 2025-11-21
>
> We thank the reviewer for their encouraging comments and insightful questions. Below, we address each point one by one.
>
> _"The theoretical part is dense and most proofs are relegated to the appendix, which is quite long (22 pages). I did not verify the correctness of the proofs in the appendix.”_
> ---------
> We understand that some of the technical proofs in our paper are dense (in particular, those of Theorems 1 and 3). While we cannot move the entirety of the proofs to the main body due to the page limit, __we have now added proof sketches after each of our main three theorems__ (lines 345-350; 396-401; 420-426) to provide intuition on the proof techniques that we employed, along with explicit pointers to which subsection of the appendix the full proof can be found in. All new content added to the main body has currently been marked in __blue__. We hope the sketches will help with the accessibility of our theoretical results. Please let us know if any of the proof sketches are unclear; we would be happy to further elaborate.
>
> _Regarding details for the competition mentioned in line 091:_
> ---------
> As pointed out in footnote 1, we have omitted the name and year of the competition and conference to preserve the anonymity of this submission, in accordance with the ICLR 2026 Author Guide. We are worried that revealing specific details about the competition can jeopardize the anonymity of the submission, especially since the number of competitions in recent top-tier AI conferences related to rank aggregation has been limited. While we understand this may make it difficult to judge the strength of the implementation, we hope that the theoretical and experimental results provided in the paper demonstrate the strengths of our method, including in comparison to other classes of natural rule picking rules (Theorem 1.II) and to current rules being used in practice (Fig. 2(a)). Still, below we answer the questions posed by the reviewer to the best of our ability without revealing identifying information about the competition or the conference.
>
> _“What place/score did the mentioned implementation get? 1st place, 2nd place, 3rd place?”_
> ---------
> Our rule was among the four rules that were jointly announced as the “winning rules” during the award ceremony, all of which received a score within <0.3% of each other, which was significantly higher than all other competing rules. In addition to checking axiomatic properties, the competition scored rules based on a specific welfare function that mapped the rules' outputs to utilities for voters, which was announced prior to the competition. Among the four winning rules, our rule was notable in that we did not perform any tuning or optimization for this welfare function, while other methods were explicitly designed to maximize this specific function, underscoring the robustness and strong general performance of $AbC$ across diverse evaluation criteria.
>
>
> _“How many competitors were there in the competition?”_
> ---------
> To the best of our knowledge, this information is not publicly available, as the competition website throughout the duration of the competition displayed only the rules with top 10 highest scores. We have reached out to the competition organizers to request data on the number of competitors, but have not yet received a response. If our paper is accepted, we will provide a link to the competition website (which has our rule listed among the winners and all other details) in our final writeup. As pointed out in our submission, this was one of the official competitions in a recent top-tier AI conference.

---

### Meta-Review · Area_Chair_VZqo · 2026-01-06

**Summary:**

The authors address the problem of selecting an appropriate method for a rank aggregation problem, noting that different aggregation rules can yield substantially different outcomes. They propose a data-driven approach that chooses the aggregation method maximising consistency under repeated data collection. Because optimising consistency is computationally hard, they introduce an efficient sampling-based implementation. The approach satisfies desirable consistency axioms and is validated experimentally on both synthetic models and real-world datasets.

The reviewers are quite positive about the paper and only raised minor issues in their reviews, regarding presentation and interpretation of the "rules to pick a rule" approach, the scope of aggregation rules covered, and the experimental study. Despite the lack of response by the reviewers (except for one, who confirmed to be satisfied), I think the authors did a good job in addressing these points in their rebuttal. Moreover, all reviewers have already been on the positive side before the rebuttal. Overall, I would say that the paper presents an interesting idea that is worked out in a sound and thorough way.

**Reviewer Concerns:**

All concerns have been addressed.

**Reviewer Scores:**

I guess the scores would have remained like this. Maybe one 6 could have turned into an 8.

---

### Decision · Program_Chairs · 2026-01-26

Accept (Poster)